# Distributed Multi-Scale Calibration of Low-Cost Ozone Sensors in Wireless Sensor Networks

**DOI:** 10.3390/s19112503

**Published:** 2019-05-31

**Authors:** Jose M. Barcelo-Ordinas, Pau Ferrer-Cid, Jorge Garcia-Vidal, Anna Ripoll, Mar Viana

**Affiliations:** 1Universitat Politecnica de Catalunya (UPC), UPC Campus Nord, 08034 Barcelona, Spain; pauferrercid12@gmail.com (P.F.-C.); jorge@ac.upc.edu (J.G.-V.); 2Institute of Environmental Assessment and Water Research, Spanish National Research Council (IDAEA-CSIC), 08034 Barcelona, Spain; anna.ripoll@idaea.csic.es (A.R.); mar.viana@idaea.csic.es (M.V.)

**Keywords:** wireless sensor networks, low-cost sensors, calibration, error estimation, air pollution sensors

## Abstract

New advances in sensor technologies and communications in wireless sensor networks have favored the introduction of low-cost sensors for monitoring air quality applications. In this article, we present the results of the European project H2020 CAPTOR, where three testbeds with sensors were deployed to capture tropospheric ozone concentrations. One of the biggest challenges was the calibration of the sensors, as the manufacturer provides them without calibrating. Throughout the paper, we show how short-term calibration using multiple linear regression produces good calibrated data, but instead produces biases in the calculated long-term concentrations. To mitigate the bias, we propose a linear correction based on Kriging estimation of the mean and standard deviation of the long-term ozone concentrations, thus correcting the bias presented by the sensors.

## 1. Introduction

Today, sensor technology installed on wireless nodes is beginning to mature with applications in various fields including precision agriculture, air pollution, location, water flow monitoring, etc. [1,2,3,4]. Air pollution is one of the major concerns in modern society due to its economic impact and on people’s health. Air pollution is mainly monitored by reference stations deployed by government organizations. Due to the high cost of these reference stations, researchers have focused their studies on low-cost air pollution sensors for pollutants such as O${}_{3}$, CO, NO${}_{x}$, CO${}_{2}$, PM2.5, etc., whose objective is to complement the data obtained by reference stations. These low-cost devices are mounted and integrated into wireless nodes that send their sensed data to database repositories for further processing, for example to elaborate high resolution air pollution maps [5,6]. One of the criticisms of this approach [7,8], currently under frequently debate [9,10], is the lack of knowledge of the performance and reliability of the long-term results made by these low-cost sensors. The reliability of low-cost sensors is an issue that has been debated several times in the literature in different fields of application [7,8,11,12].

Several research projects have explored the possibility of deploying a low-cost wireless sensor platform to collect air quality data. Our H2020 CAPTOR project (https://www.captor-project.eu/en/) is one of them and it is based on the assumption that the combination of citizen science, collaborative networks and environmental grassroots social activism helps to raise awareness and find solutions to air pollution problems. In order to engage people and raise social awareness, the CAPTOR project (2016–2018) has developed wireless sensor nodes able to measure tropospheric ozone (O${}_{3}$). The nodes have been deployed in three testbed wireless networks in Spain, Italy and Austria. Although the development of nodes in terms of hardware, software and communications have been challenging, one of the main concerns in the development of the project is the uncertainty of the captured data. One of the causes of this low reliability is that low-cost sensors are calibrated in manufacturer controlled environments such as laboratory chambers or have not even been calibrated. In general, the ideal case would be that low-cost sensors are calibrated by manufacturers. However, this is not always the case, due to the cost of calibration and the fact that in many cases the calibration depends on the environmental conditions at the deployment site [4]. Therefore, calibration should be performed under the same environmental conditions under which the sensor will have to measure.

When calibration is performed in the field, under network deployment conditions, calibration is said to be in an uncontrolled environment [13]. In this paper, we explain how is the calibration process of about one hundred and forty sensors that measure tropospheric ozone in a network deployed in Spain, Italy and Austria in the H2020 CAPTOR project. The calibration of ozone sensors has been challenging for several reasons: (i) ozone is a seasonal pollutant, e.g., in Europe ozone concentrations are high in summer between May and September and low the rest of the year; (ii) in winter, as ozone concentrations are low, it is difficult to calibrate the sensors due to the lack of data representative of the environmental conditions present in summer; and (iii) the lifetime of low-cost sensors is short, close to a year or a year and a half, which prevents low-cost sensors from being calibrated for long periods of time with large data sets.

Most studies [14,15,16,17,18,19,20,21] are characterized by using large data sets that divide into two parts: one part to train the model called training data set and another part to validate the model called testing data set. However, all these authors do not analyze the behavior of the models when the calibration parameters have to estimate pollutant concentrations in the long-term. There is little knowledge, even using linear models, of how long-term estimation behaves. This problem is relevant in a deployment of sensors whose objective is not to investigate how a sensor performs but rather to calibrate the sensor with a small data set (calibration period) near a reference air quality monitoring station and then deploy it elsewhere. In this case the calibration parameters are used to estimate long-term pollution concentrations without being able to verify the quality and accuracy of the measurements taken.

Our paper aims to answer the question of how the location, amount of data in the training set, calibration period and environmental conditions impact on the estimation of ozone concentrations, both in the short and long term, providing an exhaustive study to calibrate ozone sensors with metal-oxide technology. Thirty-five wireless nodes with one hundred and forty ozone sensors, thirty-five temperature and relative humidity sensors deployed in the H2020 CAPTOR project have been used for this purpose [22]. Summarizing, in this paper, we:investigate the effect of the size of the data set on calibration and its ability to give short-term ozone concentrations,perform an analysis of the behavior of one hundred and forty sensors, showing that sensors of the same family behave differently with great variability in terms of Root Square Mean Error (RMSE),study the impact of calibration in different locations and the impact of different environmental conditions as well as bias over time,finally, propose a mechanism of calibration in two phases: in the first, we calibrate the sensor and we obtain data calibrated with bias. In the second phase, we correct the values with bias by means of a linear correction based on the calculation by means of a distributed network of sensors that estimate the mean and the standard deviation of the ozone concentration.

The outline of the paper is as follows. Section 2 enumerates the related work. Section 3 describes the testbeds and data sets used for the analysis. Section 4 explains the calibration method based on multiple linear regression techniques with array of sensors. Section 5 presents the results of calibrating one hundred and forty ozone sensors, including short-term and long-term calibration, and the impact of ambient conditions and bias on the sensors. Section 6 describes a distributed mechanism to correct the bias produced in the sensor. Finally, concluding remarks are made in Section 7.

## 2. Related Work

There is a rich literature on how to calibrate low cost sensors applied to wireless sensor networks (WSN). We start with two surveys [13,23]. Barcelo-Ordinas et al. [13] describe the different approaches to calibrate sensors in applications with sensors of light, temperature, relative humidity, vibration, accelerometer, localization, synchronization, target location applications or air pollution. Maag et al. [23] focuses more on characterizing how sensors are calibrated in the air pollution area. In general, the most used calibration architecture [13] defined for low-cost air pollution sensors in WSN is centralized, micro, collocated, pre/post-calibration, off-line, non-blind with an array of M sensors [14,15,21,24,25,26]. The reason for such an approach lies in the need to have a reference station that provides accurate data (micro, collocated and non-blind calibration). Wireless nodes send sensor data to a server or repository where it can be calibrated (centralized and off-line calibration). Finally, many air pollutants are directly or inversely related to other pollutants (e.g., ozone is inversely related to nitrogen oxide due to titration) or to environmental parameters (e.g., ozone correlates to ambient temperature), which means that we need an array of sensors to calibrate a specific pollutant.

Other authors follow other approaches to calibrate sensors in a distributed way. For example, Saukh et al. [27] and Maag et al. [28] propose using a multi-hop architecture instead of collocated-centralized architectures for calibrating O${}_{3}$ and NO${}_{2}$ low-cost sensors in a WSN. The authors deployed a set of wireless nodes in a mobile network, on top of buses in Zurich, and these nodes use a window interval when the vehicles pass near (opportunistically) a reference station or other already calibrated wireless node (multi-hop). Here, in this context, multi-hop calibration means that a sensor node has been calibrated using an already calibrated sensor node.

From an algorithmic point of view, many of the low cost sensors have been calibrated with linear models such as multiple linear regression [14,15,21,24,25,26]. Recently, it has been proven that some of these sensors have non-linear behaviors and has been investigated how air pollution sensors can be calibrated with non-linear methods. For example, Esposito et al. [25] analyse air pollution using support vector regression (SVR) and artificial neural networks (ANN) and compare these signal processing techniques against multiple linear regression (MLR) and Gaussian process regression (GPR) showing that when non-linearity appears in the response function, ANN and SVR outperform MLR and GPR at the cost of more computational resources. Spinelle et al. [14,15] also use ANN for comparing sensors (NO${}_{2}$, O${}_{3}$, CO and CO${}_{2}$) from several manufacturers and technologies (metal-oxide and electro-chemical) and how they behave in providing good estimates regarding reference values. Liu et al. [21] analyze arrays of air pollution sensors (CO, CH${}_{4}$, CeO${}_{2}$ and C${}_{3}$H${}_{8}$ among others) using a Bayesian approach. Other authors also study calibration using models such as k-nearest neighbors (KNN) [18,29], Gaussian processes [18], random forest (RF) [16,17,18] or support-vector regression (SVR) [17,19,20]. However, all of these papers do not investigate the performance of calibration in estimating long-term contaminant concentrations.

Among the works that investigate the impact that calibration has on long-term estimation we can find the following: Yamamoto et al. [4] shows how temperature sensors behave very differently when calibrated in a different place from where they are deployed, mainly because they have different environmental conditions of temperature and relative humidity, which impacts on the long-term prediction. Castell et al. [30] also evaluate electro-chemical sensors and how environmental conditions impact long-term predictions, showing very low accuracy in the long-term and emphasizing that the information provided by manufacturers is not sufficient for such calibration.

In addition, to mitigate the bias affecting long-term estimates, we propose an adjustment to these estimates using a linear correction based on the estimation of the mean and standard deviation of long-term concentrations. For this estimation of the mean and standard deviations we use Kriging. Kriging interpolates values using a weighted average based on points in the neighborhood of the target point. Schneider et al. [31] also use Kriging to make urban air quality maps from low-cost NO${}_{2}$ sensors in Oslo, Norway. The authors show how Kriging provides good results depending on several factors including the number of observations or the spatial distribution of nodes.

## 3. Data Set and Wireless Sensor Nodes

Gas sensors to measure gases such as CO, CO${}_{2}$, O${}_{3}$, NO, NO${}_{2}$ are sensors that follow multiple linear responses. Tropospheric ozone (O${}_{3}$) formation occurs when nitrogen oxides (NOx) and volatile organic compounds (VOCs) react in the atmosphere in the presence of sunlight. In general, to calibrate ozone sensors and depending on the type of sensor, (metal-oxide or electro-chemical), it is necessary to measure O${}_{3}$, NO${}_{2}$, temperature and relative humidity [14,15]. During the ozone measurement campaign in the H2020 CAPTOR project in the summers of 2017 and 2018, nodes have been deployed in Spain, Italy and Austria. The nodes called captor nodes were built by UPC (Universitat Politecnica de Catalunya), Barcelona, Spain, following the DIY (Do It Yourself) philosophy, uses Arduino technology with a sensor shield board that attaches four SGX Sensortech MICS 2614 metal-oxide O${}_{3}$ sensors in each captor node, a temperature (Temp) sensor and a relative humidity (RH) sensor, Figure 1a. Each captor node is powered by an external power supply and is connected to Internet using Wifi or 3G.

The data used in this paper belong to the captor nodes deployed in the summer 2017 ozone measurement campaign in Spain (Catalonia) and Italy (Piemonte, Veneto, Lombardia and Emilia Romagna). Twenty-five nodes were deployed in Spain and ten in Italy, with a total of one hundred and forty ozone sensors, thirty-five temperature sensors and thirty-five relative humidity sensors.

Captor nodes have been calibrated using reference stations, Figure 1b, in Spain and Italy, Table 1. Palau Reial reference station is an urban reference station in a large town like Barcelona, where ozone in average in summer is low. We have used Palau Reial because it is near UPC and the nodes could easily be tested in a first phase. The other reference stations are located in country-side areas where ozone in summer is high and are near the volunteer houses, Figure 1c, where the wireless sensor nodes were placed. The nodes have undergone the following calibration process:**Phase 1:** in this phase, called calibration phase, captor nodes 01–08, 10–12, 14, 15, 18–20, 22–30 and 34 have been placed, Table 2, and calibrated during three or four weeks between the months of May and June 2017 in the reference stations of Spain and Italy, Table 1, where it is known that large concentrations of ozone are produced in summer,**Phase 2:** these nodes have been placed in volunteer houses near the reference stations where the nodes were calibrated during phase 1. The nodes have remained during the months of July, August and September 2017 in the volunteer houses,**Phase 3:** in this phase the nodes that were deployed in the volunteer houses have been relocated to recalibrate for two weeks in October 2017 in the same reference stations where they were calibrated in phase 1.

The nodes 09, 13, 16, 17, 21, 31, 32, 33 and 35 have been maintained during the period that lasts the three phases in reference stations with the objective of carrying out calibration studies.

Each sample is the average of a set of multiple consecutive samples taken every five minutes. In order to avoid non-representative values due to voltage spikes, we eliminate 5% of the highest values and 5% of the lowest values. The sample is then sent every half hour using wireless communication to a database in a repository where the sensors are off-line calibrated. Finally, the nodes are deployed in the homes of volunteers in areas of high ozone concentrations. The coefficients estimated during the calibration, phase 1, are loaded into the nodes using wireless communication and from that moment on, all actual ozone concentration values measured by sensors in the volunteers’ homes are sent to the database. These concentrations can be viewed using a smart-phone app or a browser on a tablet or PC.

Two types of data sets have been used in the paper. For those nodes (01–08, 10–12, 14, 15, 18–20, 22–30 and 34) that have been deployed in volunteer houses, the data set consists of three or four weeks of calibration, between May and June 2017, depending on the node and two weeks at the end of September, which respectively correspond to the calibration periods of phases 1 and 3. For those nodes (09, 13, 16, 17, 21, 31–33 and 35) that have been deployed throughout the campaign at reference stations, the data set covers the entire campaign period, nearly 21 weeks, between May and the end of September 2017.

As can be deduced, the data of the nodes deployed in volunteer houses can only be compared with reference values during the calibration period, phases 1 and 3 and allow us to investigate the behavior of a large number of sensors in a short calibration period that corresponds to a deployment situation. On the other hand, the data of the nodes deployed during a long period in reference stations allow us to vary the period dedicated to the training of the data in the calculation of the calibration parameters and therefore will be the data used during most of the paper.

## 4. Multi-Array Non-Blind Calibration of Ozone Sensors Using Linear Models

In our first approximation and throughout the entire Section 4, we will assume that an uncalibrated sensor is placed relative to a reference station that provides reference ozone concentration values [13]. By collocated, we mean that an uncalibrated node is closely deployed in the vicinity of the reference station, i.e., less than two meters, and that the node and reference station take samples close enough in time [27]. In addition, as previously mentioned, samples are sent from the Captors nodes to a repository, so the calibration is off- line. Since the data from the reference station is obtained at a certain point and the captor nodes send data to a server, the calibration is micro and non-blind [13]. Finally, as explained in the Section 3, each Captor node includes a set of sensors, so the calibration uses an array of sensors.

For calibrating SGX Sensortech MICS 2614 metal-oxide O${}_{3}$ sensors, let us consider an array of M sensors: O${}_{3}$, temperature and relative humidity sensors (M = 3). The data set of size N is split in two parts. The training set of size N${}_{1}$ is used to obtain the calibration coefficients in the MLR model. The testing set of size N${}_{2}$ is used to validate how these calibration coefficients predict ozone concentrations. A multiple linear regression model (MLR) in multi-array calibration sensor assumes M predictors, one for each sensor of the array, taking the form of [32]:(1)$$y_{n}∼f\left(\beta ,x_{n}\right)=\beta_{0}+\sum_{j=1}^{M}\beta_{j}x_{nj}+\epsilon_{n}\;\;\;\;\;\;\;\;\;n=1,...,N_{1}$$
where $\epsilon_{n}$ is a random error term, Gaussian distributed with zero mean and variance $\sigma^{2}$. The model assumes that *y* is a vector of *N*${}_{1}$ snapshots (samples) with the ground-truth or reference values, and $x_{j}$ (*j* = 1, ..., *M*) are vectors of size *N*${}_{1}$ with the data measured by each of the *M* sensors of the array with the uncalibrated or raw values. For each of the ozone sensors s${}_{k}$ (*k* = 1, 2, 3, 4) used at each node, the resulted model will be:(2)$$y_{n}∼f\left(\beta ,x_{n}\right)=\beta_{0}+\beta_{1}x_{n,s_{k}}+\beta_{2}x_{n,Temp}+\beta_{3}x_{n,RH}+\epsilon_{n}\;\;\;\;\;\;\;\;\;n=1,...,N_{1}$$

Thus, MLR assumes a data set of *N*${}_{1}$ snapshots that are used to derive the coefficient estimates ${\hat{\beta }}_{j}$. These coefficient estimates will allow to calculate in what we call a **single-scale (SS)** mechanism new calibrated values:(3)$${\hat{y}}_{n}={\hat{\beta }}_{0}+\sum_{j=1}^{M}{\hat{\beta }}_{j}x_{nj}\;\;\;\;\;\;\;\;\;n=N_{1}+1,\cdots ,N$$

The parameters $\hat{\beta_{j}}$ are estimated by minimizing Equation (Equation 4), the residual sum of squares (RSS):(4)$$RSS=E\left(\beta \right)=\sum_{n=1}^{N_{1}}{\left(y_{n}-f\left(\beta ,x_{n}\right)\right)}^{2}=\sum_{n=1}^{N_{1}}{\left(y_{n}-\beta_{0}-\sum_{j=1}^{M}\beta_{j}x_{nj}\right)}^{2}$$

Calling ${\hat{y}}_{n}$ = f($\hat{\beta }$,x${}_{n}$), the mean square error (MSE) measures the average of the squares of the errors. The MSE is the second moment (about the origin) of the error, and thus incorporates the variance of the calibration curve. However, most of the times it is more interesting to compare different sizes of data sets. root mean-squared error (Restricting N to *N*${}_{1}$ or *N*${}_{2}$, we can calculate the RMSE for the training or for the testing set.) (RMSE) is measured in the same scale than the target value $y_{n}$:(5)$$RMSE=\sqrt{MSE}=\sqrt{\frac{1}{N}\sum_{n=1}^{N}{\left(y_{n}-{\hat{y}}_{n}\right)}^{2}}$$

The error is positive except whenever the function f($\hat{\beta }$,$x_{k}$) passes exactly through each target point $y_{n}$ in which case the error will be zero and the sensor will be perfectly calibrated. Moreover, the Coefficient of Determination (*R*${}^{2}$) measures the proportion of variability in Y that can be explained using X and it is bounded between 0 and 1. When *R*${}^{2}$ is close to 1 indicates that a large proportion of the variability in the response has been explained by the regression. Given that $\mu_{y}$ = $\frac{1}{N}\sum_{n=1}^{N}y_{n}$ is the mean of the reference data:(6)$$R^{2}=1-\frac{\sum_{n=1}^{N}{\left(y_{n}-{\hat{y}}_{n}\right)}^{2}}{\sum_{n=1}^{N}{\left(y_{n}-\mu_{y}\right)}^{2}}$$

Each ozone sensor s${}_{k}$ (k = 1, 2, 3, 4) at each node is calibrated individually. For this purpose, Equation (Equation 2) is used, in which for each sensor s${}_{k}$ (k = 1, 2, 3, 4) of the node, the raw data of the ozone sensor and the data of temperature and relative humidity are used. In the H2020 project, for each node, the sensor with the lowest test RMSE value was chosen to represent the node.

When evaluating long-term RMSE, it is interesting to break it down into two components: the mean bias and the centred RMSE. The mean bias measures the difference between the mean of the predictions $\mu_{\hat{y}}$=$\frac{1}{N}\sum_{n=1}^{N}{\hat{y}}_{n}$ and the mean of the reference values $\mu_{y}$.

(7)$$Bias=\frac{1}{N}\sum_{n=1}^{N}\left({\hat{y}}_{n}-y_{n}\right)=\mu_{\hat{y}}-\mu_{y}$$

The centred RMSE (CRMSE) centers the measured and reference values with respect to their respective averages.

(8)$$CRMSE=\sqrt{\frac{1}{N}\sum_{n=1}^{N}{\left[\left({\hat{y}}_{n}-\mu_{\hat{y}}\right)-\left(y_{n}-\mu_{y}\right)\right]}^{2}}$$

It can be easily proven that:(9)$$RMSE^{2}=CRMSE^{2}+Bias^{2}$$

And finally, when studying the long-term we can normalize the metrics with respect the reference data standard deviation $\sigma_{y}$: RMSE/$\sigma_{y}$, CRMSE/$\sigma_{y}$ and Bias/$\sigma_{y}$.

## 5. Experimental Evaluation

In our wireless sensor network, the sensors are calibrated for three weeks, phase 1, and then deployed in volunteer houses, phase 2. This calibration is evaluated in the next section.

### 5.1. Sensor Family Analysis

We first investigate the difference in performance of the family of one hundred and thirty-six low-cost metal-oxide sensors deployed in the CAPTOR project. All sensors come from the same manufacturer and should behave similarly according to the datasheet. The data set used in this section corresponds to the data of all captor nodes. For the nodes that were deployed in the volunteer houses, Table 2, we have taken the data for phase 1. For the nodes that were all the campaign deployed in reference stations, we have taken the data for three weeks in June. To validate the calibration process, the data for the three weeks has been divided into 65% for the training set and 35% for the test set.

We can see in the Figure 2a the high variability in the RMSE of the different sensors ranging from 24.0 to 7.8 μg/m${}^{3}$. To install one or another sensor of the same family impacts on the quality of the data obtained irrespective of the calibration process [26]. Figure 2b displays the RMSE for each sensor against *R*${}^{2}$ (Coefficient of Determination). We can observe that low RMSE corresponds to *R*${}^{2}$ high, although some high RMSE values also had a high *R*${}^{2}$. There were a 76.5% of the sensors that had *R*${}^{2}$ values greater than 0.75, which shows that multiple linear regression is an acceptable model for this type of sensors.

Figure 3a,b illustrate how locations under different environmental conditions show similar performance. Sensors calibrated at the same location also show high variability in calibration performance. A good solution, given the low price of the sensors, is for each node to install several metal- oxide ozone sensors. In this way, the redundancy of the sensors adds reliability at the same time that it allows to choose the best representative value of the ozone concentration. Figure 4a,b display the RMSE for the four sensors of each node of the testbeds of Spain and Italy respectively. Each Captor node is labeled from 1 to 25 (Spanish testbed) and from 1 to 9 (Italian testbed) [26].

In the y-coordinates, the RMSE for the entire data set is displayed for each of the four sensors of each Captor node, labeled s${}_{1}$, s${}_{2}$, s${}_{3}$ and s${}_{4}$. You can see how the sensor with the best (minimum) RMSE marked with the largest circle in the Figure 4 can be chosen to estimate ozone concentration. As an example, the node 17012 placed in Tona and labeled as 10 in the Figure 4a has sensors with testing RMSE of s${}_{1}$ = 8.3, s${}_{2}$ = 17.7, s${}_{3}$ = 13.3 and s${}_{4}$ = 14.6 μg/m${}^{3}$ with *R*${}^{2}$ equal to 0.94, 0.75, 0.86 and 0.83 respectively. The sensor chosen to calculate node 17012 is s${}_{1}$. In the event of s${}_{1}$ sensor failure, the Captor node offers resilience by predicting with s${}_{3}$ sensor as the second best RMSE of the four ozone sensors.

### 5.2. Calibration of Sensors with Different Amounts of Data

We then validate the use of the multiple linear regression model and analyze whether this model correctly gives concentrations of O${}_{3}$ in the short-term and in the long-term. As mentioned in Section 4, the data set is divided into two parts, one for the training of the coefficients and the other for the estimation (testing) of the data. When the estimation is close, a few weeks, to the training data set, we call it short-term estimation. When the estimation is distant, several weeks, from the training data set, we call it long-term estimation. In general, a good calibration should be able to maintain similar errors in short- and long-term predictions unless the sensor has a bias, drift, or ages.

To study short and long-term estimations, we chose the nodes that have been placed throughout the campaign in the reference stations, Table 2: nodes 17013 (Manlleu), 17016 (Vic) and 17017 (Tona) of the Spanish testbed. Each of these captor nodes has four O${}_{3}$ sensors labeled s${}_{1}$, s${}_{2}$, s${}_{3}$ and s${}_{4}$. These nodes, Figure 5, have been placed in reference stations from May 8th to October 4th 2017 (21 weeks) with 30-min time resolution. Manlleu, Vic and Tona are areas with high concentrations of ozone. Tona is 62.5 km North of Barcelona, Spain. In a line that goes South to North, Vic is 10.5 km from Tona, and Manlleu is 13.3 km from Vic. Figure 6, Figure 7 and Figure 8 show the different cases studied. A period [t${}_{k}$,t${}_{k+1}$] represents a fixed interval of time, e.g., a day or a week, where data is analyzed.

**Case 1 (Figure 6):** the test data set was fixed from week fourteen, starting in August 7th , to week seventeen, ending in September 2nd . The training data set increased from week one to week thirteen (*forward direction*). The objective was to see the effect of increasing the training data set when the training data set approaches the test data set.**Case 2 (Figure 7):** the test data set was fixed from week fourteen, starting in August 7th, to week seventeen, ending in September 2nd. The training data set increased from week thirteen to week one (*backward direction*). The goal was to see the effect of increasing the training data set when the training data set moves away from the test data set.**Case 3 (Figure 8):** the training data set was fixed from week one, starting in May 28th, to week three, ending in June 19th. The test data set was fixed to one week, but moves from week four to week twenty-one (forward direction). The goal was to observe the effect of giving short- term and long-term ozone concentrations. This case was representative of an actual deployment, where sensors were calibrated for a limited period of time [t${}_{0}$,t${}_{wt}$] in a specific location and then deployed in a different place but close to where they have been calibrated.Figure 9a plots the RMSE as a function of the training set size (forward direction case) for two sensors on Captor nodes 17013, 17016 and 17017 placed at the Manlleu, Vic and Tona reference stations. Average daily concentrations at these stations ranged approximately between 50–110 μgr/m${}^{3}$, Figure 10b. Moreover, instantaneous ozone concentrations for one thousand samples (twenty days) are shown in Figure 11a,b.

For one or two week training data set sizes the RMSE is small, Figure 9a. Then, the RMSE increases for training data sets of three to four weeks and then decreases the RMSE as the training data set grows. In general, the larger the data set, the better the RMSE, but as seen in Figure 6, this is because the training data set approaches the test data set and the environmental conditions of the test data set are similar to those of the training data set. If we analyze Figure 9b, where we start from a small data set close in time to the test data set (backward direction case), it can be observed that after two or three weeks of training data set the RMSE does not improve. We conclude that for short-term estimates a training set of between two and four weeks is sufficient, and for long-term estimates we need large training set sizes.

However, in a deployment of a network of wireless air pollution sensors, the sensors are calibrated over a short period of time, e.g., a few weeks, and then placed in the chosen locations. The key question is to learn how ozone concentrations can be calculated over long-term based on calibration coefficients pre-calculated during short training phases. To verify short or long-term outcomes, case 3, Figure 10a plots the results when the test data set moves away from the training data set. The training data set has been chosen with a size of three weeks. Figure 10a shows the RMSE measured by days. Each tic of the x-axis represents a day starting on the day following the completion of the training data set. The figure shows Captor nodes 17013, 17016 and 17017 placed in the reference stations in order to compare the measurements with reference data. Only the sensor s${}_{4}$ is shown in the figure. We can observe in Figure 10a that the RMSE of the test set is not constant and oscillates every day from values that sometimes reach about 40–45 μg/m${}^{3}$ and others are not more than 5–8 μg/m${}^{3}$. Figure 10b shows the average ozone concentration per day taken at the reference stations. The figure displays that the ozone concentration also oscillates and that they are seasonal, with lower averages at the end of summer, the month of September. The conclusion is that long-term estimates are worse than short-term estimates when there is a small training data set size.

Table 3 shows the training and test RMSE and *R*${}^{2}$ for one hundred days of measurements. The training set size is of three weeks and the test set size is of fifteen weeks. We can observe:there was a large difference between sensors performance, even being in the same place and under the same environmental conditions. For example, the four sensors in Captor 17017, had testing RMSE oscillating between 13.7 μg/m${}^{3}$ and 22.1 μg/m${}^{3}$,even having days in which the RMSE was large, the testing RMSE in some cases are in the same order of magnitude than the training RMSE, as can be seen in sensors s${}_{1}$ of Captor node 17013 or sensor s${}_{1}$ of Captor node 17017. On the other hand, other sensors such as s${}_{2}$ of Captor node 17016 has testing RMSE of 35.4 μg/m${}^{3}$, that is larger than the training RMSE, 11.8 μg/m${}^{3}$. Captor node 17016 has worst RMSE, probably caused by high peaks of ozone concentration in Vic area on weeks 3, 4 and 5, Figure 11b,the high variability of the RMSE observed in Figure 10a can be explained observing Figure 11a,b, which show the instantaneous testing estimates of the ozone concentration in Captors 17013 (Manlleu) and 17016 (Vic). Each sample represents half an hour and therefore, the 1000 samples represented about 20–21 days. Each cycle was a day, as ozone reached higher peaks during the day and its lower concentrations at night. We can observe that low-cost metal-oxide sensors have difficulty in reaching high values of ozone concentration (larger than 150 μg/m${}^{3}$) and low values of ozone concentration (lower than 30 μg/m${}^{3}$). This situation means that those days with very high or very low ozone concentration increase the RMSE.

### 5.3. Study of the Sensor Bias

To better understand the long-term fluctuations of the RMSE in Captor nodes, Figure 12a,b show the decomposition of the normalized RMSE into Bias/$\sigma_{y}$ and into CRMSE/$\sigma_{y}$ (variance) over time of sensor s${}_{4}$ at Captor node 17013 (Manlleu) and sensor s${}_{4}$ at Captor node 17017 (Tona). Each unit on the x-axis is one day.

It can be observed that both nodes have biases and a high variance over time. The reason for these behaviors, and which have a strong impact on the RMSE obtained, is the reaction that measurements have to different environmental conditions, e.g., temperature and relative humidity, making it difficult to estimate long-term ozone concentrations. This impact of environmental conditions has been observed in other sensors. Yamamoto et al. [4] analyze the impact of the calibration site and environmental conditions on temperature sensors, and Castell et al. [30] study the impact of temperature and relative humidity on sensors with electro-chemical technology.

From now on, we will call this type of calibration *single-scale calibration* because we only use the pre-calculated calibration coefficients and we have not performed any kind of bias correction and we will use the acronym SS to indicate it.

### 5.4. Impact of the Environmental Conditions in the Calibration

We have observed in the previous sections that the RMSE reported by the sensors have a high variability and that the sensors had biases due to environmental conditions. To verify the impact of environmental conditions on long-term estimations, we have measured in Figure 13a the RMSE as a function of the average day-to-day temperature, and in Figure 13b, the RMSE as a function of the average day-to-day relative humidity.

The green dots represent the s${}_{1}$ sensor of the Captor 17013 node placed in the Manlleu reference station in case of single-scale calibration. The multi-scale case, dots in orange, will be explained in Section 6. It can be seen that high temperatures and low relative humidity produce high RMSE values for the single-scale calibration case. Low temperature and high relative humidity values, on the other hand, produced low RMSE values, indicating better calibration. This explains the bias shown in the previous section, i.e., if we calibrate under ambient conditions different from those that occur in the period when estimations of ozone concentrations are made, bias will appear.

Figure 14a illustrates the mean temperature as a function of relative humidity in Captor node 17013 by showing that high temperatures correspond to days with low relative humidity values. Figure 14b draws the RMSE as a function of the reference ozone concentration obtained by the day-to-day reference station. As in the previous figure, the green dots show the RMSE for the single-case calibration case. The figure confirms that high RMSE values generally correspond to dry days with high temperatures and low relative humidity values. Under these conditions, sensors with metal-oxide technology behave worse than when there are low temperatures and wet days.

## 6. Distributed Multi-Scale Calibration

As mentioned in the previous section, a challenge in calibrating an actual deployment in air pollution networks is that given a fixed data set of a few weeks, we will be able to provide calibrated ozone concentrations, i.e., good estimations of the test data set, regardless of time separation from the training data set. In this section we present a calibration model that we call *multi-scale* (MS) that corrects the bias produced in the ozone concentrations calibrated with respect to the model used in the previous section and that we called *single-scale* (SS).

### 6.1. Multi-Scale Calibration Using Multiple Linear Regression Correction with Exact Values

We call single-scale estimation (SS) to, ${\hat{y}}_{n}^{SS}$, the instant estimation on the test data set, i.e., using Equation (Equation 3). Now, we call multi-scale estimation (MS), ${\hat{y}}_{n}^{MS}$, the corrected estimation on the value calculated by the single-scale model.

(10)$${\hat{y}}_{n}^{MS}=a_{1}+a_{2}*{\hat{y}}_{n}^{SS}\;\;\;\;\;\;\;\;\;n=N_{1}+1,\cdots ,N.$$

Now, taking expectation and variance in Equation (Equation 10), we obtain expressions for constants *a*${}_{1}$ and *a*${}_{2}$:(11)$$a_{2}=\frac{\sigma^{MS}}{\sigma^{SS}.}$$
and
(12)$$a_{1}=\mu^{MS}-a_{2}*\mu^{SS}=\mu^{MS}-\frac{\sigma^{MS}}{\sigma^{SS}}*\mu^{SS}.$$
where $\mu^{SS}$ and $\sigma^{SS}$ are the mean and standard deviation of the instantaneous testing data for the single-scale model, ${\hat{y}}_{n}^{SS}$, and $\mu^{MS}$ and $\sigma^{MS}$ are the real mean and standard deviation of ozone concentrations in the period of the testing data for the multi-scale model, ${\hat{y}}_{n}^{MS}$.

Moreover, reformulating Equations (Equation 7) and (Equation 8) applying the linear correction and normalizing with respect to the reference standard deviation $\sigma_{y}$ we obtain:(13)$$\frac{Bias^{MS}}{\sigma_{y}}=\frac{\left(\mu_{\hat{y}}^{MS}-\mu_{y}\right)}{\sigma_{y}}$$
and,
(14)$$\frac{CRMSE^{MS}}{\sigma_{y}}=\sqrt{\frac{1}{N}\sum_{n=1}^{N}{\left[\frac{\left({\hat{y}}_{n}^{MS}-\mu_{\hat{y}}^{MS}\right)}{\sigma_{y}}-\frac{\left(y_{n}-\mu_{y}\right)}{\sigma_{y}}\right]}^{2}}=\sqrt{\frac{1}{N}\sum_{n=1}^{N}{\left[\frac{\frac{\sigma_{\hat{y}}^{MS}}{\sigma_{\hat{y}}^{SS}}\left({\hat{y}}_{n}^{SS}-\mu_{\hat{y}}^{SS}\right)}{\sigma_{y}}-\frac{\left(y_{n}-\mu_{y}\right)}{\sigma_{y}}\right]}^{2}}$$

As it can be seen in Equations (Equation 10)–(Equation 12), to calculate ${\hat{y}}_{n}^{MS}$ you need to know $\mu^{SS}$, $\mu^{MS}$, $\sigma^{SS}$ and $\sigma^{MS}$. To estimate $\mu^{SS}$, $\mu^{MS}$, $\sigma^{SS}$ and $\sigma^{MS}$ a K-sample window has been taken. From now on we set this K window to the equivalent size in samples of a day or a week (multi-scale with daily/weekly averages correction). Then, the multi-scale calibration process consists of the following stages:sensors placed during phase 1: obtain single scale calibration parameters (Equations (Equation 1) to (Equation 4)) from sensors using the MLR model,sensors placed during phase 2: take the latest K estimates obtained by the single scale method ([${\hat{y}}_{n-K}^{SS}$, ${\hat{y}}_{n}^{SS}$]) using Equation (Equation 3), and calculate $\mu^{SS}$ and $\sigma^{SS}$,sensors placed during phase 2: take a K-size window and estimate $\mu^{MS}$ and $\sigma^{MS}$,sensors placed during phase 2: estimate ozone concentrations (${\hat{y}}_{n}^{MS}$) using the multi-scale correction, Equation (Equation 10).

The biggest difficulty in applying the multi-scale correction is in estimating $\mu^{MS}$ and $\sigma^{MS}$ (stage 3), since $\mu^{SS}$ and $\sigma^{SS}$ can be obtained from ${\hat{y}}_{n}^{SS}$. Before proposing a way to estimate $\mu^{MS}$ and $\sigma^{MS}$ let us check that if we knew the exact values of $\mu^{MS}$ and $\sigma^{MS}$, the linear correction improves the bias of the instant estimates. To do this, we are going to use the values of the mean and standard deviation of the ozone concentration values provided by the reference station when the node is placed with respect to that station. For this, we use the same data set as in Section 5.2: Captor nodes 17013 (Manlleu), 17016 (Vic) and 17017 (Tona).

In the case of multi-scale calibration with exact correction, $\mu_{\hat{y}}^{MS}$ = $\mu_{y}$ and $\sigma_{\hat{y}}^{MS}$ = $\sigma_{y}$ (mean and standard deviation of the reference station data in the window interval [n−K, n]). Thus, Bias${}^{MS}$ = 0 and:(15)$$\frac{CRMSE^{MS}}{\sigma_{y}}=\sqrt{\frac{1}{N}\sum_{n=1}^{N}{\left[\frac{\frac{\sigma_{\hat{y}}^{MS}}{\sigma_{\hat{y}}^{SS}}\left({\hat{y}}_{n}^{SS}-\mu_{\hat{y}}^{SS}\right)}{\sigma_{y}}-\frac{\left(y_{n}-\mu_{y}\right)}{\sigma_{y}}\right]}^{2}}=\sqrt{\frac{1}{N}\sum_{n=1}^{N}{\left[\frac{\left({\hat{y}}_{n}^{SS}-\mu_{\hat{y}}^{SS}\right)}{\sigma_{\hat{y}}^{SS}}-\frac{\left(y_{n}-\mu_{y}\right)}{\sigma_{y}}\right]}^{2}}$$

Figure 15a,b reproduces the results of Figure 10a (Section 5.2, case 3), where the RMSE of the test set is drawn day by day for the whole summer campaign. In this Figure 15, it can be observed that for the Captor nodes 17016 (Vic) and 17017 (Tona), the average per day is more stable, still with some peaks due to environmental conditions, but not with as high variability as observed in the single-scale case.

Figure 16a,b shows the mean normalized bias${}^{MS}$ and the normalized CRMSE${}^{MS}$ for the Captor 17016 and 17017 nodes. It can be seen that the mean normalized bias is zero and the normalized CRMSE${}^{MS}$ decreases with respect to the single-scale case (green) when exact correction (black) is used.

In addition, Table 4 shows the values of the ozone concentrations before applying the correction, testing RMSE (SS), and after applying the correction, testing RMSE (MS), during the one hundred days of the campaign. For the correction we have considered two cases: (i) correction with mean and standard deviation with one day size K-window; and (ii) correction with mean and standard deviation with one week size K-window. It can be observed that the average RMSE decreases in all sensors when the values are daily, with reductions ranging from 25% to 75%. When the values are weekly, the reduction is smaller, although still considerable. The reason is that the estimation of the mean value and standard deviation of the ozone concentration is less accurate. These results show that it is possible to correct the bias presented by the sensors if we are able to find estimators of the mean and standard deviation. However, finding mechanisms that are capable of calculating such estimators is difficult, and if that is not possible, single-scale calibration is the best thing that can be done using MLR.

One of the effects of multi-scale correction is to mitigate the impact of environmental conditions, i.e., temperature and relative humidity, on the RMSE. By looking again at Figure 13a,b, we can compare the effect of temperature and relative humidity in the RMSE for the single-scale case (green dots) and the multi-scale case (orange dots). You can see how the correction of mean and standard deviation softens the RMSE in all environmental conditions. In addition, we can observe that the reduction in the RMSE is greater when the RMSE is large, while the reduction is smaller when the RMSE is small. Examples are the reduction of an RMSE in the range of 28–34 μg/m${}^{3}$ to values in the range of 13–17 μg/m${}^{3}$, while RMSE values in the range of 10–14 μg/m${}^{3}$ are reduced to values of 7–10 μg/m${}^{3}$.

### 6.2. Multi-Scale Estimation Using Kriging

As the monitoring area is suburban/rural and the nodes have been deployed near reference stations, we propose to estimate the mean and standard deviation to correct the values obtained by the single-scale model using Kriging. Ozone concentrations in suburban/rural areas are known to show a relatively low spatial variability due to the secondary nature of this pollutant, which means that the Kriging approach could be adequate for this pollutant and type of area [33]. This would not be the case for other pollutants such as NO${}_{2}$ in urban areas, because of the high spatial variability of NO${}_{2}$ emissions and ambient concentrations [34]. In urban areas it would be necessary to use more sophisticated estimation models that would take into account the propagation of the pollutant in such areas.

Kriging is a technique used in geostatistical models that estimates the value of a function at a given point computing the weighted average of the known values of the function in the vicinity of the point. Although Kriging is not intended as a distributed mechanism, it allows estimating and therefore calibrating parameters at a localized point. In its distributed version, the nodes can gossip (Gossip is a type of routing mechanism for WSN that allows information to be communicated between nodes.) their values together with their GPS locations to the nodes participating in the Kriging mechanism. Kriging is closely related to regression analysis and is a particular case of the Gaussian process model [35], where the domain is space. Kriging estimators interpolate the value of the random field at an unobserved location from observations of its values at nearby locations. The objective is to estimate the value at a localized point by a weighted average of the neighboring points to the location of the estimated target. In general, uniformly distributed dense data locations give good estimates, while sparse or clustered data locations can provide worse estimates.

In the CAPTOR project the set of neighbours is composed of other sensors already calibrated and reference stations. Since ozone is not necessarily constant and the neighboring nodes can be in an area of a few kilometers, we chose Universal Kriging with a Gaussian semivariogram model as an estimator of the mean $\mu^{MS-Kriging}$ and the standard deviation $\sigma^{MS-Kriging}$ of ozone concentrations. We want to emphasize that we do not use Kriging to estimate the instantaneous value of ozone concentrations, but it is for the means and standard deviations that are used to correct the estimates of the single-scale model. Therefore, we estimate Kriging on the mean and standard deviation of 2-hop sensors using neighboring sensors that can be either pre-calibrated sensors or reference stations. Finally, we corrected the instantaneous ozone concentration values using Equation (Equation 10), where a${}_{1}$ and a${}_{2}$, Equations (Equation 11) and (Equation 12), are obtained using the Kriging estimates $\mu^{MS-Kriging}$ and $\sigma^{MS-Kriging}$.

To validate multi-scale calibration using linear correction with Kriging estimation, we calculated RMSE and *R*${}^{2}$ for target nodes 17013, 17016 and 17017. We use the same data set as in the previous section. As before, the training data set is three weeks. The test data set consists of one hundred days. Since these nodes are placed in reference stations, for the validation to be fair, we have to be careful that the reference stations where the target node is positioned do not participate in the Kriging process. All reference stations and Captor nodes in the Spanish geographical area participate in the final calibration of all nodes except baseline nodes 17013, 17016 and 17017 which have only been used to validate the model.

Table 5, Table 6 and Table 7 show the process of estimating the means and standard deviations for each of the three target nodes 17013, 17016 and 17017, using Kriging estimates. As an example, take node 17013, located in the Manlleu reference station, so this reference station does not participate in the algorithm in this case. To estimate its mean and standard deviation, the distributed algorithm consists of five steps, Table 5: (1) we obtain the mean value and the standard deviation of the ozone concentrations in the Vic and Tona reference stations for the time window [n-K,n] and calculate with Kriging the value of $\mu^{MS}$ and $\sigma^{MS}$ in the coordinates of nodes 17012 and 17023; (2) repeat the process in the coordinates of the node 17027, but now in the Kriging process participate the mean value and the standard deviation of the stations of Tona and Vic and the nodes 17012 and 17023; (3,4) the process is repeated with the rest of the nodes that participate in the calibration of the objective node 17013, and finally, in step (5) you get $\mu^{MS}$ and $\sigma^{MS}$ in the coordinates of the objective node 17013 and you apply the multi-scale calibration process explained in Section 6.1.

Looking at Table 4, we can see: (i) the improvement using the multi-scale model with Kriging estimation with respect to the single-scale case; and (ii) how good is the multi-scale model with Kriging estimation with respect to the baseline estimation with exact values of the Section 6.1. We can observe that the MS model with Kriging estimation has better performance in all sensors of node 17016 (Vic) for both daily and weekly averages and standard deviations. In the single-scale case, the lowest RMSE was 18.9 μg/m${}^{3}$ for the sensor s${}_{4}$ and the highest was 35.4 μg/m${}^{3}$ for s${}_{2}$. With the estimate of Kriging MS, the RMSE for s${}_{4}$ goes down to 11.2 μg/m${}^{3}$ (daily correction) and to 13.4 μg/m${}^{3}$ (weekly correction).

The values are worse than in the baseline case using the Vic reference station but close, showing an improvement over the single-scale case. In the case of node 17013 placed in the Manlleu reference station, the multi-scale model with mean correction and standard deviation with Kriging estimators improve the single-scale case, although not as much as in node 17016. The reason is that this node already had better RMSE values in the single-scale case than the 17016 node and therefore there is not as much room for improvement. Finally, for the 17017 node placed in the Tona reference station, the multi-scale model with daily or weekly averages correction with Kriging estimates works the same or even worse than the single-scale.

The case of nodes 17013 (Manlleu) and 17017 (Tona) are different from the case of node 17016 (Vic). The reason is that since node 17016 is in the center of the area studied, Figure 5, two reference stations (Manlleu and Tona) participate in the estimation of the mean and standard deviation with Kriging. On the other hand, to calibrate node 17013, we only used one reference station (Vic), since we have excluded the Manlleu station for the validation of the results. The same occurs with the node 17017 in which we only used one reference station (Vic) in the calibration process.

To better understand the causes of these results, Figure 17a,b and Figure 18a,b draw the daily estimates of the mean and standard deviation using Kriging estimators on nodes 17016 and 17017. We have chosen the s${}_{4}$ sensor of node 17016 because the performance improvement between single-scale and multi-scale is big, and the s${}_{4}$ sensor of node 17017 that has a worse performance in multi-scale compared to single-scale. One can observe the plot for the mean and standard deviation of the ozone concentration measured by: (i) the reference stations (black), (ii) the single-scale model (green), and (iii) the multi-scale models with Kriging estimators (orange). We can see that for the estimation of $\mu^{MS-Kriging}$ and $\sigma^{MS-Kriging}$, the sensor s${}_{4}$ of the node 17016 estimates with little error the mean and the standard deviation, while the sensor s${}_{4}$ of the node 17017 estimates with little error the standard deviation but underestimates the mean. The cause can be explained from Figure 10b, where it is observed that the ozone concentration measurement in Tona is higher than in Vic and Manlleu. So, doing a Kriging using these stations that have lower averages underestimates the average. This shows the importance of having reference stations close to the nodes involved in the Kriging process for good results in estimating the mean and standard deviation.

In Figure 19, the average daily RMSE is drawn for the single-scale case (green), multi-scale with correction with exact values (black) and multi-scale with correction with Kriging estimators (orange) in the s${}_{4}$ sensors of nodes 17016 and 17017. Observing the sensor s${}_{4}$ of the node 17016—similar results were obtained in the other sensors of the nodes 17013 and 17016—it can be observed how the multi-scale model with correction with Kriging estimators has higher RMSE than with the correction with exact values but less than with the single-scale. This happens when the correction is made with daily and weekly values. On the other hand, for the reasons explained above, the sensors of node 17017 in general give worse performance in the multi-scale due to the underestimation of the mean in the Kriging method.

We can finally compare in Figure 16a,b the mean bias for the Captor 17016 and 17017 nodes when using the single-scale and multi-scale case with exact values and with Kriging estimates of the mean and standard deviation. It is observed that for node 17016 the mean bias and the variance decrease, while for node 17017 the variance decreases slightly but underestimates the mean bias because it only uses one reference station, Vic, too far away and with a mean ozone concentration lower than in Tona.

The conclusions that can be drawn from this validation is as follows: (i) if there are no reference stations involved in the Kriging estimation process, it is best to use the single-scale case; (ii) if reference stations are available, an estimation of the mean and standard deviation and therefore a multi-scale correction can be made, provided that these reference stations have means of similar ozone concentrations and are close to the target node; (iii) if there are reference stations which, although close, have a very different mean ozone concentration, e.g., above or below, the Kriging estimate will overestimate or underestimate ozone concentrations, so in this case, it is best to use the single-scale model.

## 7. Conclusions

In this paper, we have evaluated the calibration process of a low-cost sensor deployment performed during the H2020 CAPTOR project in Spain, Italy and Austria. Thirty-five Captor nodes with one hundred and forty ozone sensors, thirty-five temperature and relative humidity sensors have been deployed. In many papers, calibration is investigated using large data sets to verify the performance of low-cost sensors. However, in an actual deployment, there are only a few weeks to calibrate the sensors. With these few data in the training data set, calibration coefficients must be obtained that are capable of estimating long-term ozone concentrations with low error. In this paper we have shown the difficulty of this task and how biases appear in the long-term estimation, mainly due to the different environmental conditions with respect to those that existed during the training.

To correct the bias presented in the long-term estimate, we propose a linear correction on the estimated results obtained using multiple linear regression calibration, which we have called single-scale. We call this correction multi-scale calibration because it uses the mean and standard deviation calculated over a period of time, e.g., a day or a week. We show that if we had the exact mean and standard deviation values, the bias is corrected in the estimation of long-term instantaneous concentrations.

However, it is difficult to find mechanisms capable of estimating such mean and standard deviations from daily or weekly values. In our case, as the nodes were deployed in non-urban areas and near reference stations, we have used a geostatistical method of interpolation called Kriging to estimate the mean and standard deviation in time intervals that allow correcting the bias and variance. In other situations, such as contaminants in urban areas, it will be necessary to use other more sophisticated estimation models. Thus, in the paper, we show how this estimation improves long-term ozone concentration values when there are reference stations close to the sensor to be calibrated.

Although in this paper we have used Kriging as a technique to interpolate the mean value and standard deviation of the linear corrector, we believe there are other techniques that can be used. The first question to mention is that we have used Kriging with a contaminant in a non-urban area. In this case, a network of calibrated sensors or reference stations the Kriging may be sufficient. In the case of other pollutants or urban areas, other interpolation techniques will have to be investigated to estimate the mean and standard deviation.

It is also necessary to investigate other techniques, not necessarily spatial, to estimate the mean and standard deviation. For example, the use of time series or if there are sufficient training data in various environmental conditions, could allow the estimation of the mean and standard deviation of ozone concentration in the long term. Finally, the issue of long-term concentration estimates is an understudied issue that is important in deployments of air pollution sensor networks. We believe that more experiments with little training data and the development of algorithms to estimate long-term pollutant concentrations are necessary.

## Figures and Tables

**Figure 1 sensors-19-02503-f001:**
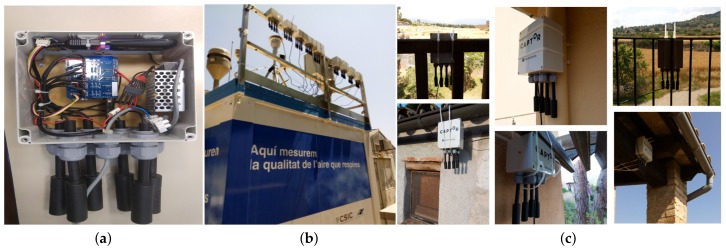
(**a**) Captor box, (**b**) Palau Reial reference station with captor nodes in calibration process, (**c**) captor nodes in Volunteer houses.

**Figure 2 sensors-19-02503-f002:**
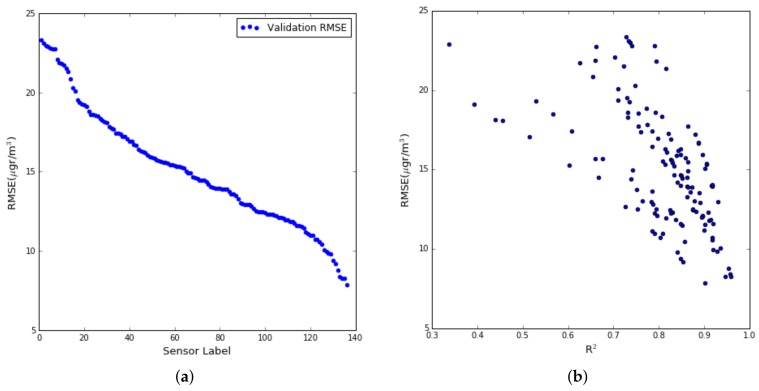
(**a**) RMSE for the set of sensors, (**b**) RMSE versus correlation coefficient (*R*${}^{2}$).

**Figure 3 sensors-19-02503-f003:**
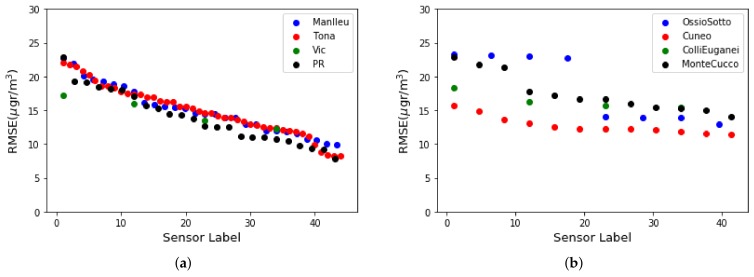
RMSE classified for testbed and place: (**a**) Spanish Testbed, (**b**) Italian Testbed.

**Figure 4 sensors-19-02503-f004:**
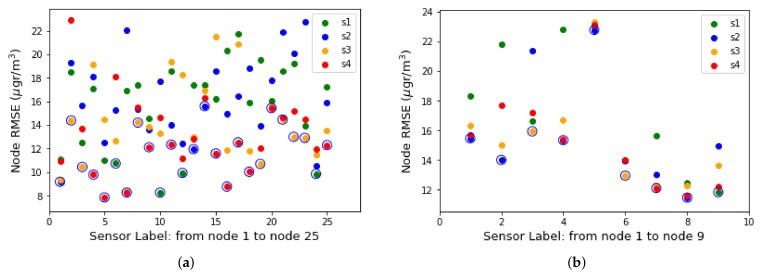
RMSE classified by testbed and node: (**a**) Spanish Testbed, (**b**) Italian Testbed.

**Figure 5 sensors-19-02503-f005:**
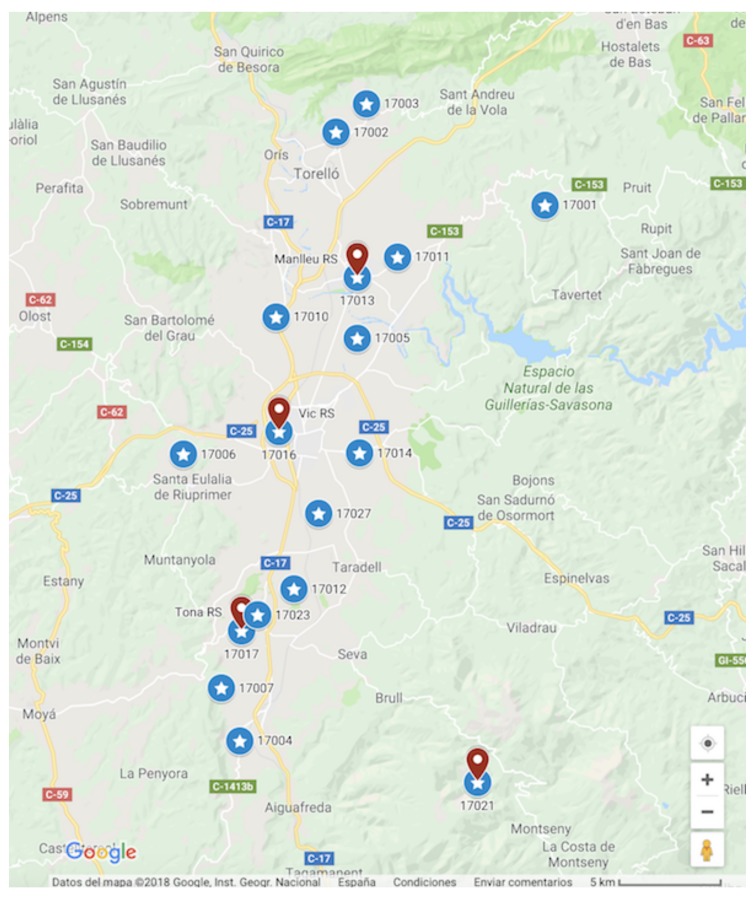
Map with the Captor nodes: the reference stations (in red), Captor nodes (in blue), Captor label number below or behind the Captor mark.

**Figure 6 sensors-19-02503-f006:**
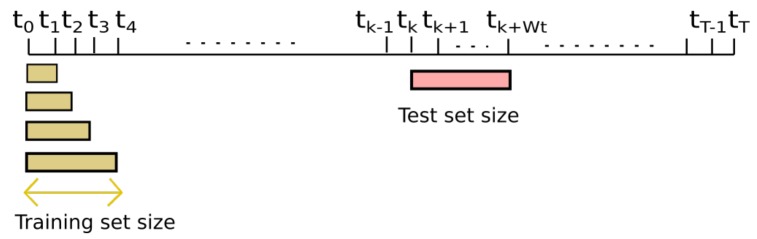
Case 1: Training set size increases in forward direction. Test set size fixed.

**Figure 7 sensors-19-02503-f007:**
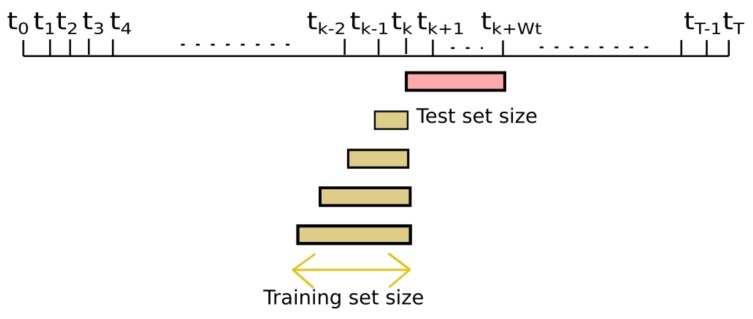
Case 2: Training set size increases in backward direction. Test set size fixed.

**Figure 8 sensors-19-02503-f008:**
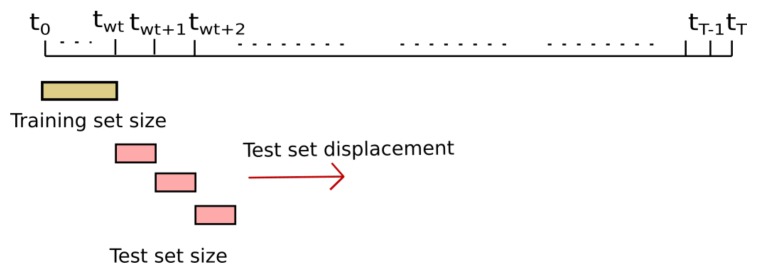
Case 3: Training set size fixed. Test set size fixed but moves in forward direction.

**Figure 9 sensors-19-02503-f009:**
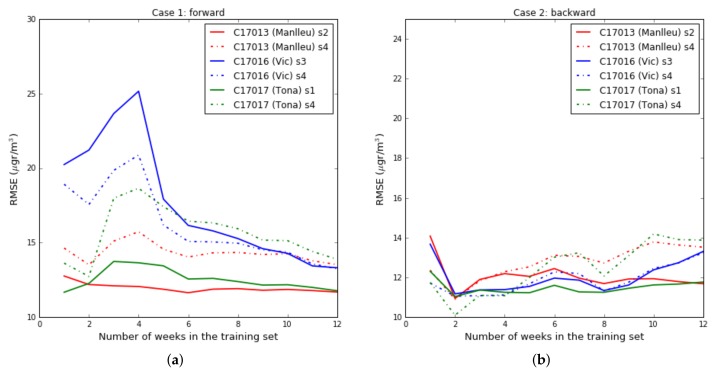
(**a**) Case 1: Training data increases in forward direction, (**b**) Case 2: Training data increases in backward direction.

**Figure 10 sensors-19-02503-f010:**
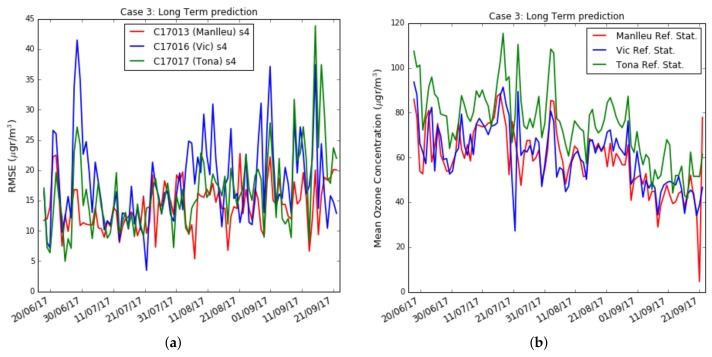
(**a**) Case 3: Captors Test RMSE in per day basis with training set of 3 weeks, (**b**) Case 3: Average ozone (μg/m${}^{3}$) in per day basis in the reference station.

**Figure 11 sensors-19-02503-f011:**
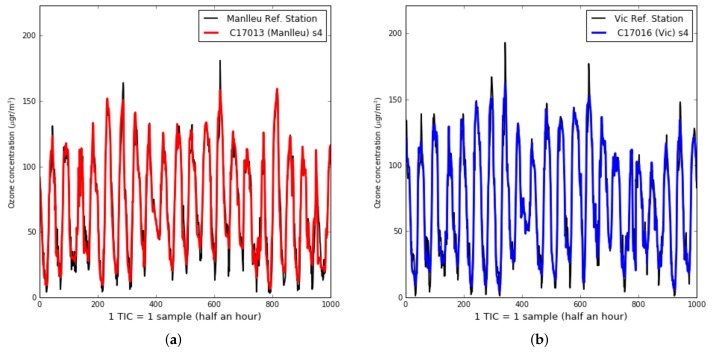
Case 3: Ozone concentration (μg/m${}^{3}$) for (**a**) Captor node C17013 (Manlleu), (**b**) Captor node C17016 (Vic).

**Figure 12 sensors-19-02503-f012:**
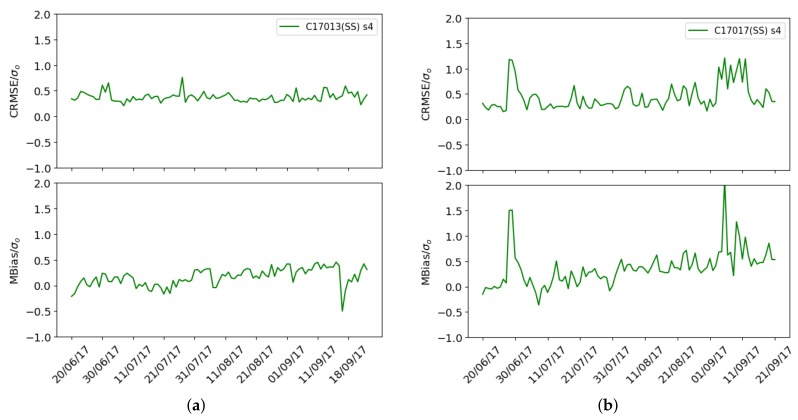
Mean normalized bias and normalized centred CRMSE: (**a**) sensor s${}_{4}$ Captor 17013 (Manlleu), (**b**) sensor s${}_{4}$ Captor 17017 (Tona).

**Figure 13 sensors-19-02503-f013:**
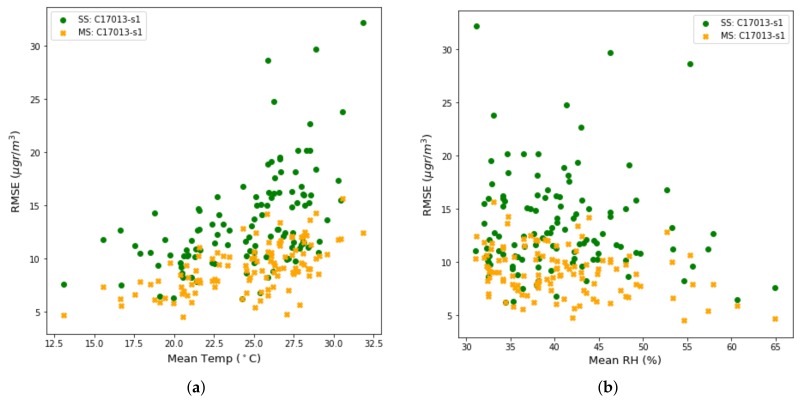
(**a**) Test RMSE versus average temperature in a per day basis, (**b**) Test RMSE versus average Relative Humidity in a per day basis. In green the single-scale case and in orange the multi-scale case.

**Figure 14 sensors-19-02503-f014:**
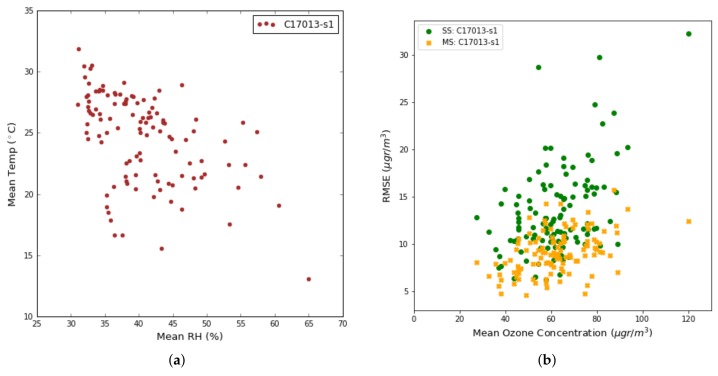
(**a**) Average temperature versus average relative humidity in a per day basis, (**b**) Test RMSE versus average ozone concentrations in a per day basis.

**Figure 15 sensors-19-02503-f015:**
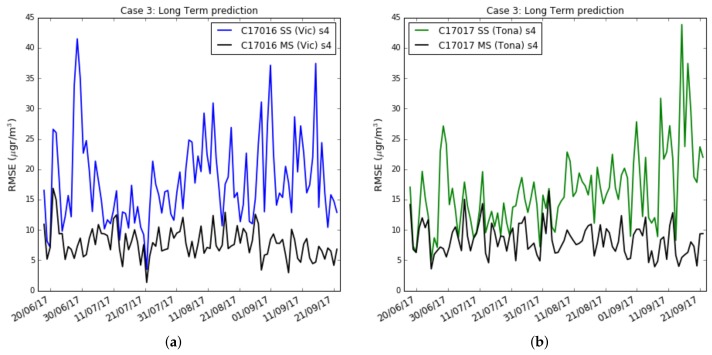
Case 3: Test RMSE in per day basis with training set of 3 weeks, single-scale versus multi-scale with exact correction, (**a**) Captor 17016 (Vic), (**b**) Captor 17017 (Tona).

**Figure 16 sensors-19-02503-f016:**
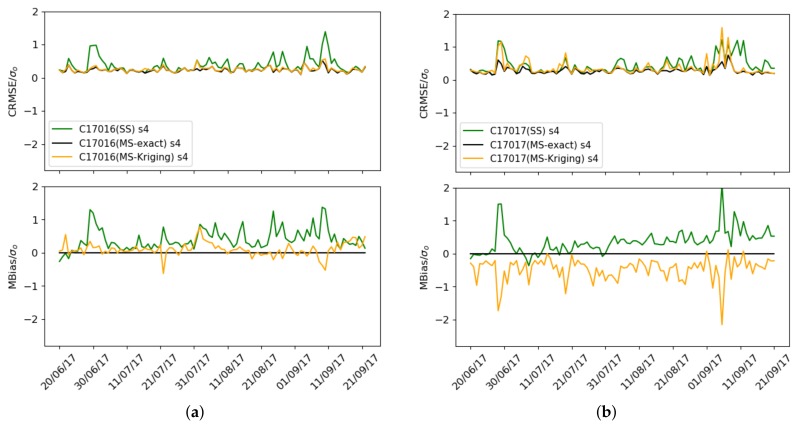
Case 3: Mean normalized bias and normalized CRMSE, single-scale (green), multi-scale with exact correction (black) and multi-scale with Kriging estimates (orange), (**a**) Captor 17016 (Vic), (**b**) Captor 17017 (Tona).

**Figure 17 sensors-19-02503-f017:**
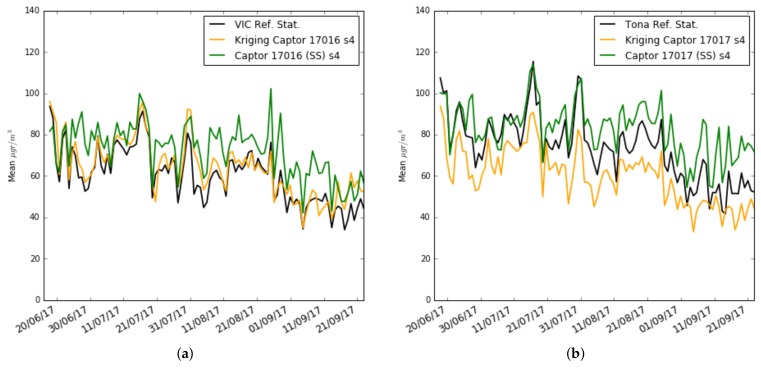
Case 3: Training set of 3 weeks: Mean ozone concentration for single-scale versus multi-scale with exact correction and Kriging estimation (daily), (**a**) in Captor 17016 (Vic), (**b**) in Captor 17017 (Tona).

**Figure 18 sensors-19-02503-f018:**
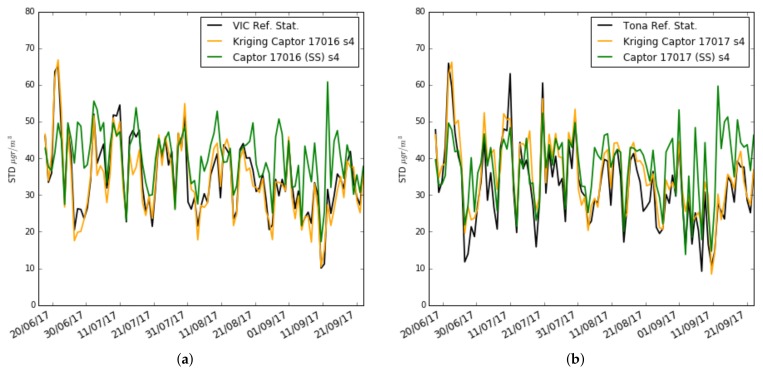
Case 3: Training set of 3 weeks: Standard deviation (STD) ozone concentration for single-scale versus multi-scale with exact correction and Kriging estimation (daily), (**a**) in Captor 17016 (Vic), (**b**) in Captor 17017 (Tona).

**Figure 19 sensors-19-02503-f019:**
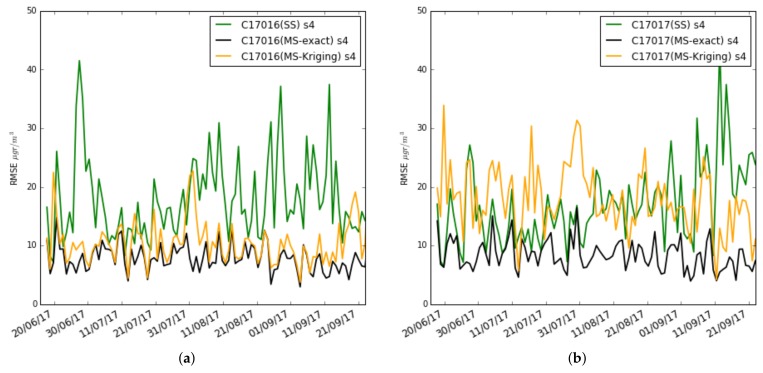
Case 3: Test RMSE with training set of 3 weeks, single-scale versus multi-scale with exact correction and Kriging estimation, (**a**) daily in s${}_{4}$ of Captor 17016 (Vic), (**b**) daily in s${}_{4}$ of Captor 17017 (Tona).

**Table 1 sensors-19-02503-t001:** Reference Stations geolocalization.

Spanish Testbed	Palau Reial	Manlleu	Tona	Vic	Montseny
	(Barcelona)	(Catalonia)	(Catalonia)	(Catalonia)	(Catalonia)
**Latitude**	41${}^{\circ }$23’14” N	42${}^{\circ }$0’6.966” N	41${}^{\circ }$50’49.7796” N	41${}^{\circ }$56’08.4” N	41${}^{\circ }$46’45.6” N
**Longitude**	2${}^{\circ }$6’56” E	2${}^{\circ }$17’13.7868” E	2${}^{\circ }$13’14.7864” E	2${}^{\circ }$14’18.8” E	2${}^{\circ }$21’28.9” E
**Italian Testbed**	**Cuneo**	**Ossio Sotto**	**MonteCucco**	**Colli Euganei**	
	**(Piemonte)**	**(Lombardia)**	**(Emilia Romagna)**	**(Veneto)**	
**Latitude**	44${}^{\circ }$22’53.6” N	45${}^{\circ }$37’14.1” N	45${}^{\circ }$02’18.8” N	45${}^{\circ }$17’21.76” N	
**Longitude**	7${}^{\circ }$32’18.4” E	9${}^{\circ }$36’41.6” E	9${}^{\circ }$40’09.7” E	11${}^{\circ }$38’32.43” E	

**Table 2 sensors-19-02503-t002:** Captor nodes history: Palau Real, Manlleu, Tona, Vic and Montseny are reference stations in Spain; Cuneo, Ossio Soto, Monte Cucco and Colli Euganei are reference stations in Italy. Captor nodes are labeled from 17001 to 17035 (abbreviated as 01 to 35). Node 17032, placed in Italy in Colli Euganei Rf. St., had a hardware failure and finally did not participate in the campaign.

Phase	Palau	Manlleu	Tona	Vic	Mont-	Cuneo	Ossio-	Monte-	Colli-	Volun-
	Reial	Rf. St.	Rf. St.	Rf. St.	seny	Rf. St.	Sotto	Cucco	Euganei	teer
	Rf. St.				Rf. St.		Rf. St.	Rf. St.	Rf. St.	Houses
**1**	09,15,	01–03,05,	04,06–07,	16		24,29	08,35	28,	30	
	18–20,	10–11,13	12,14,17,			31		33–34		
	26		21–23,							
			25,27							
**2**	09	13	17	16	21	31	35	33	32	01–08,
										10–12,
										14–15,
										18–20,
										22–30,
										34
**3**	09,15,	01–03,05,	04,06–07,	16		24,29	08,35	28,	30	
	18–20,	10–11,13	12,14,17,			31		33–34		
	26		21–23,							
			25,27							

**Table 3 sensors-19-02503-t003:** Training and testing RMSE, *R*${}^{2}$ for nodes 17013, 17016 and 17017.

	17013 (Manlleu)				17016 (Vic)				17017 (Tona)			
	s1	s2	s3	s4	s1	s2	s3	s4	s1	s2	s3	s4
**Training RMSE (** μ **g/m** ${}^{3}$ **)**	12.7	12.3	18.7	12.1	18.5	11.8	10.9	10.9	12.0	12.9	16.5	11.8
**Training *R*^2^**	0.91	0.91	0.81	0.92	0.81	0.92	0.93	0.93	0.91	0.90	0.84	0.91
**Testing RMSE (** μ **g/m** ${}^{3}$ **)**	15.1	17.0	20.5	15.6	23.1	35.4	23.4	18.9	13.7	22.1	21.1	17.6
**Testing *R*^2^**	0.85	0.81	0.72	0.84	0.64	0.17	0.63	0.76	0.87	0.65	0.68	0.78

**Table 4 sensors-19-02503-t004:** Training RMSE, R2 and total testing average RMSE for nodes 17013, 17016 and 17017.

	17013 (Manlleu)				17016 (Vic)				17017 (Tona)			
	s1	s2	s3	s4	s1	s2	s3	s4	s1	s2	s3	s4
**Single-Scale (SS)**												
**Training RMSE (** μ **g/m** ${}^{3}$ **)**	12.7	12.3	18.7	12.1	18.5	11.8	10.9	10.9	12.0	12.9	16.5	11.8
**Training *R*^2^**	0.91	0.91	0.81	0.92	0.81	0.92	0.93	0.93	0.91	0.90	0.84	0.91
**Testing RMSE (** μ **g/m** ${}^{3}$ **)**	15.1	17.0	20.5	15.6	23.1	35.4	23.4	18.9	13.7	22.1	21.1	17.6
**Testing *R*^2^**	0.85	0.81	0.72	0.84	0.64	0.17	0.63	0.76	0.87	0.65	0.68	0.78
**Multi-Scale (MS) with daily averages correction with exact values**												
**Testing RMSE (** μ **g/m** ${}^{3}$ **)**	11.1	12.8	15.3	11.1	11.6	8.3	7.6	8.1	10.3	10.8	15.4	8.5
**Testing *R*^2^**	0.92	0.89	0.84	0.92	0.91	0.95	0.96	0.96	0.92	0.92	0.83	0.95
**Multi-Scale (MS) with weekly averages correction with exact values**												
**Testing RMSE (** μ **g/m** ${}^{3}$ **)**	12.9	14.8	18.4	12.6	13.7	12.7	10.8	10.6	11.8	12.6	17.0	10.2
**Testing *R*^2^**	0.76	0.85	0.78	0.89	0.87	0.89	0.92	0.92	0.90	0.89	0.79	0.93
**Multi-Scale (MS) with daily averages correction with Kriging estimates**												
**Testing RMSE (** μ **g/m** ${}^{3}$ **)**	14.0	15.4	17.5	14.0	14.0	11.4	10.9	11.2	19.1	19.4	22.5	18.1
**Testing *R*^2^**	0.87	0.84	0.80	0.87	0.87	0.91	0.92	0.92	0.74	0.73	0.64	0.77
**Multi-Scale (MS) with weekly averages correction with Kriging estimates**												
**Testing RMSE (** μ **g/m** ${}^{3}$ **)**	14.8	16.5	19.7	14.7	15.9	15.1	13.6	13.4	20.5	21.0	24.0	19.5
**Testing *R*^2^**	0.86	0.82	0.74	0.86	0.83	0.85	0.88	0.88	0.70	0.69	0.59	0.73

**Table 5 sensors-19-02503-t005:** Multi-scale calibration with Kriging estimates (Captor node 17013).

Calibration of Node 17013 Collocated with Manlleu Ref. Stat.		
Step #	Nodes to be Calibrated	Nodes Participating in the Kriging Process
1	17012, 17023	Vic and Tona Ref. Stations
2	17027	Vic and Tona Ref. Stations and nodes 17012, 17023
3	17006, 17014	Vic and Tona Ref. Stations and nodes 17012, 17023, 17027
4	17005, 17010	Vic Ref. Station and nodes 17006, 17014, 17027
5	test node 17013	Vic Ref. Station and nodes 17005, 17010

**Table 6 sensors-19-02503-t006:** Multi-scale calibration with Kriging estimates (Captor node 17013).

Calibration of Node 17016 Collocated with Vic Ref. Stat.		
Step #	Nodes to be Calibrated	Nodes Participating in the Kriging Process
1	17005, 17010, 17012, 17023	Manlleu and Tona Ref. Stations
2	17006, 17014, 17027	Manlleu and Tona Ref. Stations and nodes 17005, 17010, 17012, 17023
3	test node 17016	Manlleu and Tona Ref. Stations and nodes 17006, 17014, 17027

**Table 7 sensors-19-02503-t007:** Multi-scale calibration with Kriging estimates (Captor node 17013).

Calibration of Node 17017 Collocated with Tona Ref. Stat.		
Step #	Nodes to be Calibrated	Nodes Participating in the Kriging process
1	17005, 17010	Manlleu and Vic Ref. Stations
2	17006, 17014	Manlleu and Vic Ref. Stations and nodes 17005, 17010
3	17027	Manlleu and Vic Ref. Stations and nodes 17005, 17006, 17010, 17014
4	17012	Vic Ref. Station and nodes 17006, 17014, 17027
5	17023	Vic Ref. Station and nodes 17006, 17012, 17014, 17027
6	test node 17017	Vic Ref. Station and nodes 17006, 17012, 17014, 17023, 17027

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
