# Peer review of "Distributed Multi-Scale Calibration of Low-Cost Ozone Sensors in Wireless Sensor Networks"

_sensors, 2019, doi:10.3390/s19112503_

Round 1

Reviewer 1 Report

Barcelo-Ordinas et al describe a method for calibrating low-cost ozone sensors. I found the paper incredibly hard to follow; a revised version needs a much clearer description of how the various calibrations were done and what data went into each calibration. For example, a key output of this analysis is development of the "multi-scale" calibration, however I cannot tell what data were used to determine this calibration.

Lines 77-78 state "many contaminants have dependencies on other contaminants. . ." Do the authors mean that certain pollutants are correlated or anti-correlated because of the chemical mechanisms governing their formation and destruction, or that low-cost sensors can be cross-sensitive to species besides the target analyte?

Section 2 seems like it should just be part of the Introduction rather than a separate section.

Page 5, description of phases and Table 1 - the naming of the phases is confusing. The "pre-calibration" phase occurs when the sensors are co-located with the reference monitors in the various cities. Presumably this co-location data is used for calibration, so calling this period "pre-calibration" is confusing. Similarly, the "post-calibration" period would make more sense if it was called "post-deployment" or something similar.

Line 176 - How were outliers identified for removal?

Figure 2 seems unnecessary. It is too small to read the details and doesn't seem to advance my understanding of the method used for calibration, which is the subject of the manuscript.

Section 4 - I am totally confused about what data are used to build the calibrations. Are all of the calibrations built from the co-location in Spain in Phase 0? Or are the calibrations built locally from co-locations in Phase 1?

Page 9 - it is not clear to me how Case 3 is fundamentally different than Case 1. In each case the calibration is deployed in the "forward" mode.

Figure 7 has "forward" misspelled.

Figures 7 and 8 show RMSE. However, the ambient O3 concentrations measured during the deployments are never shown, so I have no sense of how large the uncertainties are relative to ambient concentrations.

Equations 11-13 define the multi-scale correction. However it is not clear what is used to define the MS parameters (e.g., in Eq 12). I assume that the single scale calibration is the unique calibration for each sensor. What then is the MS calibration? What data are used to calculate mu and sigma MS?

The last section deals with a kriging method. This seems like it is only relevant for ozone, which we would expect to have regional patterns. For other pollutants with sharper spatial gradients, the kriging method is unlikely to help. Also, if part of the sensor "calibration" is to assume an overall spatial pattern of the pollutant, what is the point of having the sensor network? Why not just krig between the regulatory monitors?

Author Response

In the following, we reproduce the comments and list of proposed changes that reviewer 1 mention and we address and give an answer to each of the comments and proposed ideas.

Answer to Referee 1

1  Barcelo-Ordinas et al describe a method for calibrating low-cost ozone sensors. I found the paper incredibly hard to follow; a revised version needs a much clearer description of how the various calibrations were done and what data went into each calibration. For example, a key output of this analysis is development of the "multi-scale" calibration, however I cannot tell what data were used to determine this calibration.

There are two types of data: (i) nodes 01-08, 10-12, 14, 15, 18-20, 22-30 and 34 that in phase 1 were placed in a reference station to be calibrated for about three weeks, in phase 2 were deployed in volunteer houses to measure ozone, and in phase 3 were recalibrated to check if they continued to behave as expected, and (ii) nodes 9, 13, 16, 17, 21,31, 32, 33 and 35 that were always placed in reference stations, table 1, and therefore have been in phase 1, 2 and 3 at the same site.

We understand that the reader is not clear about these phases and the data sets they produced. Therefore, a) we have clarified the description of the data sets in section 3, lines 172-184, and b) we have described at each section which data set is used, e.g., section 5.1. Sensor family analysis (lines 235-239), section 5.2 Calibration of sensors with different amounts of data (lines 270-276), section 6.1. Multi-Scale calibration using Multiple Linear Regression correction with exact values (lines 411-412), section 6.2. Multi-scale estimation using Kriging (lines 476-477).

2. Lines 77-78 state "many contaminants have dependencies on other contaminants..." Do the authors mean that certain pollutants are correlated or anti-correlated because of the chemical mechanisms governing their formation and destruction, or that low-cost sensors can be cross-sensitive to species besides the target analysed?

Indeed, we were referring to the inter-dependencies between certain pollutants and environmental parameters. These include, for example, the anti-correlation between ozone and NO concentrations due to titration, or the correlation between ozone and ambient temperature. This is now explained in the text (lines 92-95) as follows:

"Finally, many air pollutants are directly or inversely related to other pollutants (e.g., ozone is inversely related to nitrogen oxide due to titration) or to environmental parameters (e.g. ozone correlates to ambient temperature), which means that we need an array of sensors to calibrate a specific pollutant."

3. Section 2 seems like it should just be part of the Introduction rather than a separate section.

We do agree with the reviewer that there are parts of the "Related work" that may be in the "Introduction".  We have removed these parts from the "Related work" section, and added them to the "Introduction" (lines 50-64). We believe the rest of section 2 does belong to a "Related work" section.

4.  Page 5, description of phases and Table 1 - the naming of the phases is confusing. The "pre-calibration" phase occurs when the sensors are co-located with the reference monitors in the various cities. Presumably this co-location data is used for calibration, so calling this period "pre-calibration" is confusing. Similarly, the "post-calibration" period would make more sense if it was called "post-deployment" or something similar.

We understand what the reviewer explains. It is true that in the deployment we use a technique that we called pre-post calibration, where in the pre-calibration phase we calibrated the sensors and in the post-calibration phase we verified calibrations of the last weeks and recalibrated. But from the paper point of view, it would be clearer to talk about a calibration phase and a recalibration phase. We have changed the text (lines 154 and 161) in this sense calling "calibration phase" to phase 1 and "recalibration phase" to phase 3. We have also removed phase 0 from Table 1 (now Table 2), see comment 7.

5.  Line 176 - How were outliers identified for removal?

What we do is to eliminate 5% of the highest values and 5% of the lowest values in order to avoid non-representative values due to voltage spikes. We have called outliers to these values that we eliminated, but our definition was not clear. We have replaced the text that said "Each sample is the average of a set of multiple consecutive samples taken every five minutes and with the outliers eliminated" with the following text (lines 164-166):

"Each sample is the average of a set of multiple consecutive samples taken every five minutes. In order to avoid non-representative values due to voltage spikes, we eliminate 5% of the highest values and 5% of the lowest values".

6.  Figure 2 seems unnecessary. It is too small to read the details and doesn't seem to advance my understanding of the method used for calibration, which is the subject of the manuscript.

Yes, we understand your point and we agree. It was informative with respect the whole architecture presented in the paper showing the final calibrated values but it does not give information on how the sensors are calibrated. We have remove it.

7.  Section 4 - I am totally confused about what data are used to build the calibrations. Are all of the calibrations built from the co-location in Spain in Phase 0? Or are the calibrations built locally from co-locations in Phase 1?

Calibration was performed during phase 1 at the reference stations near the site where the nodes were deployed in the volunteers' homes. Phase 0 is a phase where we calibrate locally to check that the sensors are functioning.

We agree with the reviewer that this process can confuse the reader and their contribution is more related to the deployment process than to the calibration itself. Therefore, we believe it is best to remove phase 0 from the text and Table 2, so as not to confuse the reader.

Finally, in section 3 (lines 172-184), we explain better the data sets used.

8.  Page 9 - it is not clear to me how Case 3 is fundamentally different than Case 1. In each case the calibration is deployed in the "forward" mode.

Case 1 and Case 3 are both in forward direction, but the difference is that in case 1 the training size increases and the test size is fixed and in case 3 the training size is fixed and the test size moves.

Case 1 shows what happens when we increase the training data set and approach the test data set. Case 2 shows what happens when we increase the training data set and move away from the test data set. In this way we study the impact of having large training sets, estimating ozone concentrations near or far from that training data set.

On the other hand, case 3 shows a closer realisation to reality. Data are taken for a few weeks (training set), the sensor is calibrated (thus the calibration parameters are obtained), and now ozone concentrations  that are increasingly far from the training data set are estimated with these parameters. Therefore, this case allows to study the calibration of data in the long-term. Long-term estimation, given a small and fixed data set, is an issue that has been little studied in the literature, and is really the challenge to solve if we deploy a sensor network.

Then, case 1 and case 3 are not the same, even if they are both in "forward" mode.

9.  Figure 7.a) has "forward" misspelled.

Thanks, the typo was in the top of the figure (now it is Figure 9.a)). We have proofread it.

10.  Figures 7 and 8 show RMSE. However, the ambient O3 concentrations measured during the deployments are never shown, so I have no sense of how large the uncertainties are relative to ambient concentrations.

Remembering that we have removed Figure 2 as suggested in your comment 6, and reorganising subsection 5.2 that now is subsection 5.1, the figures mentioned by the reviewer now are labeled as Figure 9 and 10.

It is correct that the absolute concentrations are not shown in Figure 9, which makes it impossible to estimate the relative weight of the RMSE. The average daily ozone concentration values are shown in Figure 10.b), ranging from approximately 50-110 μgr/m3. In addition, an example of instantaneous ozone concentrations is shown in Figures 11(a) and 11(b). In this Figure 11, it can be seen that the maximum ozone concentrations at the sensor node can reach 180 μgr/m3. This is now clarified in the text (lines 292-296) as follows:

"Figure 9.a) plots the RMSE as a function of the training set size (forward direction case) for two sensors on Captor nodes 17013, 17016 and 17017 placed at the Manlleu, Vic and Tona reference stations. Average daily concentrations at these stations ranged approximately between 50-110 μgr/m3 (Figure 10.b). Moreover, instantaneous ozone concentrations for one thousand samples (twenty days) are shown in Figure 11.a) and 11.b)."

In addition, figures 9, 10 and 11 are on the same page making it easier to understand the results.

11.  Equations 11-13 define the multi-scale correction. However it is not clear what is used to define the MS parameters (e.g., in Eq 12). I assume that the single scale calibration is the unique calibration for each sensor. What then is the MS calibration? What data are used to calculate mu and sigma MS?

We have eliminated eq (1) and (2) as suggested by reviewer 2, and included a new eq (2) to clarify the calibration model, thus, eq (11-13) are now eq (10-12). We have clarified in section 6 (lines 395-405) how multi-scale calibration works. We have explained the steps to follow to perform this calibration including what data are used to estimate μSS, μMS, σSS and σMS: multi-scale correction with exact values (lines 413-414) and with Kriging estimates (lines 487-494).

12. The last section deals with a kriging method. This seems like it is only relevant for ozone, which we would expect to have regional patterns. For other pollutants with sharper spatial gradients, the kriging method is unlikely to help. Also, if part of the sensor "calibration" is to assume an overall spatial pattern of the pollutant, what is the point of having the sensor network? Why not just krig between the regulatory monitors?

 Kriging is considered a good approach for ozone concentrations due to the relatively low spatial variability of this pollutant in rural areas, when compared to more variable pollutants such as PM or NO2. The value of having a sensor network is that it increases the spatial density of the air pollutant data, which is too low when only reference stations are considered. Adding sensor data to the reference data sets, for a kriging analysis, increases the resolution of the resulting model and in addition the sensors provide additional points for model validation, which would only be 5 (in our study area) if only reference stations were used. Therefore, using the sensor network improved the robustness of the kriging analysis.

We hope that all the questions posed by the reviewer have been answered satisfactorily.
Sincerely,

Associate Prof. Jose M. Barcelo-Ordinas
MsC. Pau Ferrer-Cid
Prof. Jorge Garcia-Vidal
Dr. Anna Ripoll
Dr. Mar Viana

Reviewer 2 Report

This article reports on a method used to correct the bias introduced on long term in the estimation of ozone concentration from a sensor network. This method is based on a geostatistical method (Krigging) used to estimate the mean and standard deviation of the long-term ozone concentration. Authors explain very well how the database is obtained from this sensor network and how environmental conditions affect the calibration for estimating long-term ozone concentration, but more information on the Krigging method must be given in section 6.

On another side, is it really useful to describe the mathematical part in section 4 from line 184 to 200?

From Line 272 to 274: I don't see high peaks of ozone concentration in Vic area in Figure 8b, is it a mistake in this figure?

From Line 293 to 294: authors explain that MLR is an acceptable model because R²=0.75 (or 76.5%???) but it is well known that PLS give better results because of multicollinearity that affects data obtained from the same kind of sensors.

Line 308: what is QoI?

Section 5.4 shows that temperature and relative humidity affect the quality of the estimation, so why these data are not directly include in the model?

In conclusion, I suggest to the authors to add more discussions about the different kinds of approach that can be used to compensate the bias or the drift of gas sensors in order to prove the originality of their model. May be the description of the database (section 5) can be reduced.

Authors must correct section numbers (section 4.1 without section 4.2? and section 6.2.1 without section 6.2.2). May be the number of sections can be reduced.

Author Response

In the following, we reproduce the comments and list of proposed changes that reviewer 2 mention and we address and give an answer to each of the comments and proposed ideas.

Answer to Referee 2

1. This article reports on a method used to correct the bias introduced on long term in the estimation of ozone concentration from a sensor network. This method is based on a geostatistical method (Krigging) used to estimate the mean and standard deviation of the long-term ozone concentration. Authors explain very well how the database is obtained from this sensor network and how environmental conditions affect the calibration for estimating long-term ozone concentration, but more information on the Krigging method must be given in section 6.

The Kriging process and how to calculate the different estimators are now explained in more detail. We have clarified in section 6 (lines 395-405) how multi-scale calibration works. We have explained the steps to follow to perform this calibration including what data are used to estimate μSS, μMS, σSS and σMS: multi-scale correction with exact values (lines 413-414) and with Kriging estimates (lines 487-494).

2. On another side, is it really useful to describe the mathematical part in section 4 from line 184 to 200?

This mathematical introduction was included to formalize the way air pollution nodes are calibrated in an uncontrolled environment. This means that the nodes with the sensors are placed ("collocated") in reference stations. The definition and formalisation of "collocated node" can be found in references [13,27]. We agree that this mathematical formalisation can be eliminated and it is enough to refer to the literature.

3. From Line 272 to 274: I don't see high peaks of ozone concentration in Vic area in Figure 8b, is it a mistake in this figure?

The reviewer is right. There was an error in the reference and we meant Figure 9.b). We have changed it, but, please, you now have to be careful with the new renumbering of the Figures. In the figure that you mention, now Figure 10.b), we were showing average daily ozone concentrations, thus, they are not higher than 110 μg/m3. However, if you look at Figure 11, you can observe the instantaneous ozone concentrations and see values that reach almost 180 μg/m3. We have mentioned this in lines 292-296 and corrected the lines 329-331 that you mention with the correct numbering.

4. From Line 293 to 294: authors explain that MLR is an acceptable model because R²=0.75 (or 76.5\%???) but it is well known that PLS give better results because of multicollinearity that affects data obtained from the same kind of sensors.

We have not used PLS (Partial Least Squares regression) because we are not fusing the data from the four ozone sensors in this paper. In case that we include the four ozone sensors in the model, we agree that there can be multicollinearity effects, and then, models such as PLS or make a PCA would reduce or eliminate multicollinearity. Summarizing:

yO3= β0 + β1 xs1O3β2 xTempβ3 xRH.

As you can observe, there is only one ozone sensor, one temperature sensor and one relative humidity sensor that participate as features in the MLR model. Each ozone sensor, s1O3, s2O3, s3O3 and s4O3 is calibrated independently of each other. We have included this formula now as eq (2), see (lines 204-206).

Finally, we agree that the sentence "There is a 76.5% R2 of values greater than 0.75 which shows that multiple linear regression is an acceptable model for this type of sensors" is confusing.

We have change it to: "There is a  76.5% of the sensors that have R2 values greater than 0.75, which shows that multiple linear regression is an acceptable model for this type of sensors" (lines 244-246).

5.  Line 308: what is QoI?

QoI means (Quality of Information) and refers to parameters such as RMSE, R2, mean bias, etc. We agree with the reviewer that the sentence "In case of sensor s1 failure, the Captor node offers resilience by estimating with sensor s3 as the second option with the worst QoI cost expected" is not clear and we have substituted it by the following sentence:

"In the event of s1 sensor failure, the Captor node offers resilience by predicting with s3 sensor as the second best RMSE of the four ozone sensors" (lines 259-261).

6. Section 5.4 shows that temperature and relative humidity affect the quality of the estimation, so why these data are not directly include in the model?

Sorry, but temperature and relative humidity were included in the MLR model as you can observe in the regression model shown in question 4. However, we note that this point was not clear enough in the manuscript. Therefore, we have added the following clarification in section 4 (lines 204-206):

"For each of the ozone sensors sk (k=1,2,3,4) used, the resulted model will be:

yn =  f(β,xn) = Î²0 + Î²1 xn,sk + Î²2 xn,Temp + β3 xn,RH + εn    n=1,...,N1."

7.  In conclusion, I suggest to the authors to add more discussions about the different kinds of approach that can be used to compensate the bias or the drift of gas sensors in order to prove the originality of their model. May be the description of the database (section 5) can be reduced.

We have added more text in the conclusions discussing approaches for compensating the bias or drift (lines 577-589).

8. Authors must correct section numbers (section 4.1 without section 4.2? and section 6.2.1 without section 6.2.2). May be the number of sections can be reduced.

Yes, the reviewer is right and we have removed these subsection titles. Now the text in section 4.1 directly goes in section 4, and the text in subsection 6.2.1 directly goes in section 6.2. Moreover, we have shifted section 5.1 and 5.2. In the revised paper what it was section 5.2 is now section 5.1 and viceversa. We believe that this structure helps to understand better the results.

We hope that all the questions posed by the reviewer have been answered satisfactorily.

Sincerely,

Associate Prof. Jose M. Barcelo-Ordinas
MsC. Pau Ferrer-Cid
Prof. Jorge Garcia-Vidal
Dr. Anna Ripoll
Dr. Mar Viana

Round 2

Reviewer 1 Report

Overall the revised manuscript is improved from the initial submission. The authors addressed most of my comments in the revised manuscript. I have several comments below.

Line 150 - What is UPC?

Lines 237-239 - Calibration of the sensors kept at reference sites use data for three weeks in June. Calibration of the sensors deployed at houses use data for Phase 1. But Table 2 does not show when Phase 1 occurred.

It's still a bit unclear if the MLR calibration is unique for each sensor or if the whole batch at each ref station is assigned a general calibration. I think it's the former but would be good to clarify.

Fig 12 could benefit from tighter y-axes.

Figure 14a says HR instead of RH

Line 407 has sigma^SS listed twice in a row.

I'm still not completely convinced of the value of the kriging in the multi-scale calibration. In the end, the sensor networks are often used to make maps or to estimate exposures. However line 528 notes the "importance of having reference stations close to the nodes involved in the Kriging process." If the distributed nodes only work well when they are close to reference monitors (and what counts as close?) it seems to detract from the value added by the network.

Please make sure that all Atmos Meas Tech Discuss and Atmos Chem Phys Discuss references are updated if the final papers have been accepted. I think that both refs 17 and 18 have been published in their final forms.

Author Response

Dear Associate Editor of Sensors journal (MDPI). In the following, we reproduce the comments and list of proposed changes that reviewer 1 mention in this second round and we address and give an answer to each of the comments and proposed ideas.

Answer to Referee 1 (2nd round)
Overall the revised manuscript is improved from the initial submission. The authors addressed most of my comments in the revised manuscript. I have several comments below.

1. Line 150 - What is UPC?
UPC means Universitat Politecnica de Catalunya, where the authors are from. It also is where we developed the nodes. We have added in the text the meaning of the acronym. However, we have put this in line 139 where we had previously mentioned that the nodes were built by UPC.

2. Lines 237-239 - Calibration of the sensors kept at reference sites use data for three weeks in June. Calibration of the sensors deployed at houses use data for Phase 1. But Table 2 does not show when Phase 1 occurred.
No, this is clarified in the text. Sensors kept at reference stations (nodes 09, 13, 16, 17, 21, 31, 32, 33 and 35) have not been calibrated at phase 1 (see line 164). We have collected these data for research purposes, and thus for using this data set in looking how to improve calibration processes.

Nodes in volunteer houses (nodes 01-08, 10-12, 14, 15, 18-20, 22-30) were placed, lines 154-163, in reference stations for 3 weeks between May and June (Phase 1) for calibrating, in volunteer houses (Phase 2) between July, August and September for observing ozone concentrations, and finally, these nodes were placed again in reference stations in October for being recalibrated during Phase 3. We have added in lines 155, 159 and 162 the months in which these nodes were calibrated.

As you can observe, the two data sets have been explained in lines 174-186. The first data set (nodes at volunteer houses) is used only in section 5.1 to show the results of calibration of sensors in a real project where we can not have the sensors months and months for calibrating since the ozone is seasonal. This fact is something that very few papers show. The second data set (nodes at reference stations) is used for investigating other calibration mechanisms that can help to correct the predictions in a real deployment. It's an engineering process that is useful for engineers that will have to deploy real sensor nodes in real deployments.

3. It's still a bit unclear if the MLR calibration is unique for each sensor or if the whole batch at each ref station is assigned a general calibration. I think it's the former but would be good to clarify.
Yes, in the MLR calibration only participated a single ozone sensor, with the temperature and relative humidity sensor. We had added in the previous round of revision eq (2) in line 206 to clarify this concept, and now we have explicitly added the following sentence in lines 223-226:

"Each ozone sensor sk (k=1,2,3,4) at each node is calibrated individually. For this purpose, eq (2) is used, in which for each sensor sk (k=1,2,3,4) of the node, the raw data of the ozone sensor and the data of temperature and relative humidity are used. In the H2020 project, for each node, the sensor with the lowest test RMSE value was chosen to represent the node."

The other ozone sensors were used only in case of failure of the first one and for research purposes, since we were interested in having a larger data set with more sensors to investigate the sensor behaviour.

4. Fig 12 could benefit from tighter y-axes.
We have tightened the y-axis of Figure 12.

5. Figure 14a says HR instead of RH
We have corrected the typo of figure 14a.

6. Line 407 has σSS listed twice in a row.
Yes, it should be "μSS and σSS". We have corrected the typo, now in line 413.

7. I'm still not completely convinced of the value of the kriging in the multi-scale calibration. In the end, the sensor networks are often used to make maps or to estimate exposures. However line 528 notes the "importance of having reference stations close to the nodes involved in the Kriging process." If the distributed nodes only work well when they are close to reference monitors (and what counts as close?) it seems to detract from the value added by the network.

We agree that the final purpose of a sensor network is to make maps or to estimate exposures. However, in WSN to get correct data it is in general necessary an engineering process. In the single-scale case the estimated data obtained using MLR as model will have an accuracy in terms of RMSE and R2. We can make maps or estimate exposures according to the accuracy obtained by this predicting MLR single-scale model.

What we are proposing now is that if we have some few reference stations near the sensor nodes we can obtain lower RMSE and higher R2. In the paper we show that with at least two stations in a radius of no more than 10 Km we obtain good results. With one station with less than 10 Km and one farther than 10 Km, we do not obtain better results than the single scale. With this multi-scale model, we can also make maps and estimate exposures, with better accuracy than the single-scale since we have corrected some bias present in the long-term prediction.

So, we conclude, that for ozone, in rural areas, if possible (availability of some reference stations near the sensor nodes) we can improve the single-scale model by engineering using a network of distributed sensor nodes plus few reference stations. If these stations are not available, then, we still can use the single-scale model, knowing that the RMSE obtained in the testing set underestimates the real RMSE in the long-term.

8. Please make sure that all Atmos Meas Tech Discuss and Atmos Chem Phys Discuss references are updated if the final papers have been accepted. I think that both refs 17 and 18 have been published in their final forms.

Yes, you are right. We had looked to the bibtex references using google scholar. In google scholar still appears as "Discuss", however, going directly to the Elsevier web page, the references appear as published as you mention. We have updated both in the Reference section.

We hope that all the questions posed by the reviewer have been answered satisfactorily.
Sincerely,

Associate Prof. Jose M. Barcelo-Ordinas
MsC. Pau Ferrer-Cid
Prof. Jorge Garcia-Vidal
Dr. Anna Ripoll
Dr. Mar Viana
